# RATE: Causal Explainability of Reward Models with Imperfect Counterfactuals

**David Reber** [1]  **Sean M. Richardson** [2]  **Todd Nief** [1]  **Cristina Garbacea** [3]  **Victor Veitch** [2,3]

## Abstract

Reward models are widely used as proxies for human preferences when aligning or evaluating LLMs. However, reward models are black boxes, and it is often unclear what they are *actually* rewarding. In this paper, we develop Rewrite-based Attribute Treatment Estimator (RATE) as an effective method for measuring the sensitivity of a reward model to high-level attributes of responses, such as sentiment, helpfulness, or complexity. Importantly, RATE measures the *causal* effect of an attribute on the reward. RATE uses LLMs to rewrite responses to produce imperfect counterfactual examples that can be used to measure causal effects. A key challenge is that these rewrites are imperfect in a manner that can induce substantial bias in the estimated sensitivity of the reward model to the target attribute. The core idea of RATE is to adjust for this imperfect-rewrite effect by rewriting *twice*. We establish the validity of the RATE procedure and show empirically that it is an effective estimator. Code is available at https://github.com/toddnief/RATE.

## 1. Introduction

Reward models (RMs) play a critical role in aligning large language models (LLMs) with desired behaviors. These models evaluate the quality of LLM outputs and are widely used in, e.g., post-training, inference-time sampling adjustment (through best-of-$n$ sampling), and LLM evaluation. However, reward models, typically implemented as fine-tuned LLMs, are black boxes. Accordingly, it can be difficult to understand what a reward model is actually rewarding. Despite their importance, explainability research for RMs remains underexplored (Lambert et al., 2024).

[1]Department of Computer Science, University of Chicago [2]Department of Statistics, University of Chicago [3]Data Science Institute, University of Chicago. Correspondence to: David Reber <reber@uchicago.edu>.

*Proceedings of the 42$^{nd}$ International Conference on Machine Learning*, Vancouver, Canada. PMLR 267, 2025. Copyright 2025 by the author(s).

To understand reward models, we would like to be able to measure their responsiveness to high-level attributes of responses, such as helpfulness, correctness, or sentiment. The ability to measure the responsiveness of an RM to an attribute would serve as an important diagnostic tool, and the purpose of this paper is to develop such a measurement procedure.

Naively, one might attempt to estimate RM responsiveness to an attribute by comparing the average reward assigned to responses with and without the attribute, using a labeled dataset. However, this approach is flawed: it conflates the target attribute's influence with correlations present in the evaluation data. For instance, suppose we are measuring how much an RM responds to "sentiment" and, unbeknownst to us, highly-negative samples in our evaluation data tend to have more typos and sloppy formatting. Then if we naively average the reward over the negative samples, and again over the positive samples, we'll inadvertently measure the effect of not just sentiment, but typos and formatting as well. In particular, this means we can measure a large effect of an attribute on the RM, *even if the RM is completely insensitive to that attribute*.

For measurements of attribute influence to be meaningful, they must be isolated from confounding factors and spurious correlations in the evaluation data. To address this, we propose formalizing RM explainability as the *causal effects* of attributes on the reward. Hence, the goal is to determine how a reward would change if we could modify a response to alter only the attribute of interest while holding all other factors constant. This counterfactual perspective isolates the causal relationship, making it possible to disentangle an attribute's true effect from confounding factors.

Notice that, if we had access to counterfactual pairs of responses (i.e., pairs where the *only* difference is the attribute of interest), we could estimate the target effect by simply comparing the rewards assigned to each response. A natural idea is to use LLMs to generate such counterfactual pairs by rewriting responses to change only the target attribute. If the rewrites were perfect, we could directly measure the causal effect of the attribute on the reward. However, in practice, LLMs produce imperfect rewrites, changing off-target attributes as well. These imperfections can substantially bias the estimated causal effect, as we will see in Section 5.

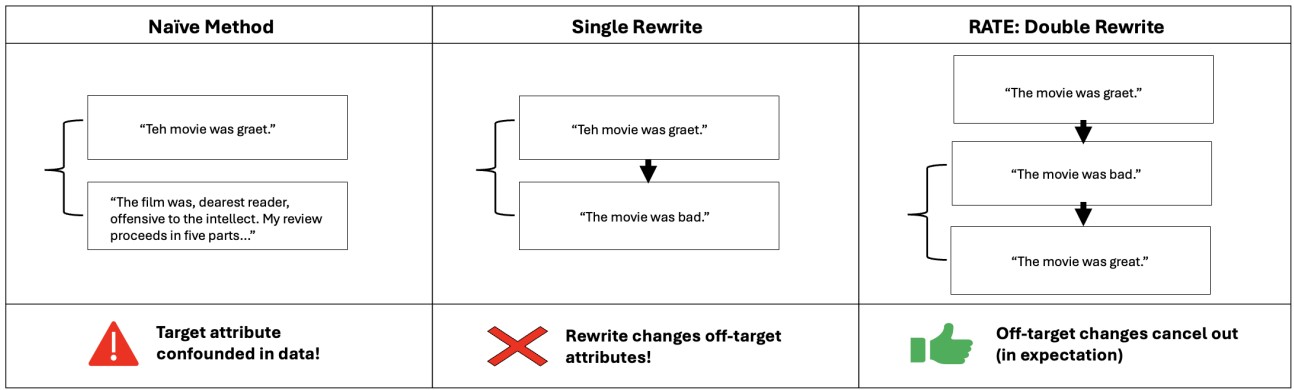

*Figure 1.* When generating counterfactual pairs, LLMs change other attributes, such as tone, length, or grammar. Empirically, using the rewrites of rewrites corrects for this bias. (Left) Naively sampling pairs which differ on the attribute of interest (e.g., sentiment) will lead to a biased estimate of the causal effect because other attributes may also change. (Middle) When we rewrite a response to change the attribute of interest (e.g., from positive to negative sentiment), the LLM may also change other attributes, such as fixing typos. (Right) Rewriting the rewritten response again tends to cancel out these off-target changes, in a manner we make precise in Section 3.

There is a growing literature on estimating the causal effects of attributes of text (Feder et al., 2022; Grimmer et al., 2022; Jin et al., 2022; Chen & Chu, 2023; Gui & Veitch, 2023). Generally, this work provide methods for estimating causal effects by either making strong assumptions on the evaluation data or compensating for potential confounding factors.

Note that our question fundamentally differs from the similarly named *counterfactual explanation* paradigm (Wachter et al., 2018) studied recently in the context of reward models by Jiang et al. (2024). While counterfactual explanations aim to explain the behavior of a model by finding a minimally perturbed input that changes the model's output, we are interested in measuring the average treatment effect of an attribute on the assigned reward, as measured across many counterfactual pairs.

The contribution of this paper is to develop a procedure that allows us to use the *imperfect* rewrites from LLMs to get a correct estimate of the causal effect of an attribute on a reward model. The development is as follows:

1. We formalize RM explainability as an Average Treatment Effect (ATE) of high-level attributes of responses, on the reward assigned to that response.

2. We develop a measurement procedure (RATE) which utilizes *two* imperfect rewrites to estimate the ATE without needing to explicitly list confounding factors in the evaluation dataset.

3. We show that RATE provides a valid estimation of the true causal effect under mild assumptions.

4. We empirically demonstrate that RATE correctly estimates causal effects, while the naive and single-rewrite

baselines are highly influenced by spurious associations. We also show that this causal-vs-correlational problem substantially affects the explanations of leading reward models using common benchmarks.

## 2. Setup

A reward function $R$ is a function that takes a prompt $x$ and a response $y$ as inputs and return a real number indicating the quality of the response for the prompt.[1] To develop an evaluation procedure, we begin with a fixed dataset of prompt-completion pairs $\{(x^i, y^i)\}$, where the $x^i$ are prompts and the $y^i$ are completions (also referred to as "responses"). We are interested in understanding how the reward model responds to a certain attribute $W$ (such as sentiment or length). We consider the case where each prompt-completion pair is labelled with $w^i = W(x^i, y^i) \in \{0, 1\}$ indicating whether the completion has the attribute of interest.[2]

We focus on binary attributes for simplicity—many attributes of interest can often be naturally binarized.

**Naive Method**   If we want to measure the sensitivity of a given reward model to an attribute of interest, such as helpfulness, the obvious approach is to take the dataset of prompt-completion pairs, label each completion as helpful

---

[1]A reward function can also be viewed as taking two responses and returning a *relative* preference between them; our results extend easily to this case as well.

[2]We include the prompt $x$ in the argument of $W$ because the attribute may be prompt-dependent. For example, $W$ might represent helpfulness, which varies based on the context given by the prompt. A recipe could be helpful for questions about cooking but not for questions about history.

or unhelpful, then check whether the rewards for the helpful responses are higher than the rewards for the unhelpful responses. Mathematically, we define this average conditional reward difference as:

$$\hat{\tau}_{\text{naive}} = \frac{1}{n_1} \sum_{(x^i, y^i): w^i = 1} R(x^i, y^i) - \frac{1}{n_0} \sum_{(x^i, y^i): w^i = 0} R(x^i, y^i)$$

where $n_1$ and $n_0$ are the numbers of examples with $W = 1$ and $W = 0$, respectively.

This estimates the correlation between the reward and $W$,

$$\mathbb{E}[R(X, Y) \mid W = 1] - \mathbb{E}[R(X, Y) \mid W = 0],$$

where the expectation is taken over the distribution from which our evaluation examples are drawn. The problem here is that, even in the infinite data limit, this quantity does not generally isolate the effect of $W$ on $R$. For instance, if the procedure we use to collect the evaluation data has a correlation between helpfulness and length, then the effect of these attributes will be conflated in the naive estimator (see Figure 1, middle). Consequently, we will misinterpret the true behavior of the reward model.

**Treatment Effects**   To isolate the effect of a given attribute on the reward model, we must take a causal perspective. Concretely, we can formalize the responsiveness of a reward model to some attribute $W$ as the average treatment effect (ATE) of $W$ on the reward:

$$\text{ATE} = \mathbb{E}[R(X, Y(1)) - R(X, Y(0))] \quad (1)$$

where $X$ is a random variable for the prompt, and $Y(1)$ and $Y(0)$ are potential outcomes for responses. This quantity is the expected change in reward if we were to change the attribute $W$ from 0 to 1, while keeping all other aspects of the response fixed. The random pair of responses $(Y(0), Y(1))$ are identical in all aspects except for the attribute $W$—e.g., if $W$ is helpfulness then each counterfactual response should have the same writing style, sentiment, topic, etc. In general, we only observe one of the counterfactual responses in our dataset (Figure 1, left)—this is the fundamental problem of causal inference (Imbens & Rubin, 2015).

**Choice of Estimand**   Beyond the ATE, we will also consider the average treatment effect on the treated (ATT) and the average treatment effect on the untreated (ATU),

$$\text{ATT} = \mathbb{E}\left[R(X, Y(1)) - R(X, Y(0)) \mid W = 1\right]$$
$$\text{ATU} = \mathbb{E}\left[R(X, Y(1)) - R(X, Y(0)) \mid W = 0\right]$$

Intuitively, if $W = 1$ is a helpful response, the ATT measures the change in reward when we take a helpful response and make it unhelpful, and the ATU measures the change

in reward when we take an unhelpful response and make it helpful. These estimands can differ substantially from each other and from the ATE (see Figure 8 in Appendix D.1). There is no reason to expect these quantities to align in general, so thought should be given to which is most relevant to the question at hand. Indeed, even human preferences are often asymmetric (Kahneman & Tversky, 2013), so we might expect reward model preferences to be as well.

## 3. Rewrite-based Attribute Treatment Estimator (RATE)

Whatever our choice of estimand, we need a method to estimate it. Here, we develop a method, RATE, that uses rewrites to estimate the causal effect of an attribute on a reward model. The core idea is to create pairs of responses where the only difference is in the attribute of interest, even if the rewrites are imperfect.

**Rewrites With LLMs**   In practice, we implement rewrites using a large language model (LLM) (see Table 1). We begin with a labeled dataset containing ground truth binary variables for attributes such as complexity, sentiment, or helpfulness. We then instruct the LLM to rewrite the responses to the opposite state of the binary variable. For example, we may instruct: "Rewrite this response to express negative sentiment and change *nothing* else."

We use $\text{Re}(x^i, y^i, w)$ to denote the rewrite operation, which takes a prompt-response pair $(x^i, y^i)$ and a desired attribute value $w$, returning a modified response $\tilde{y}^i$ such that $W(x^i, \tilde{y}^i) = w$. When $x^i$ is clear from context, we drop it for brevity and write the operation as $\text{Re}(y^i, w)$, even though the rewriter in general may depend on the input prompt.

**Rewrite Instructions**   There is significant flexibility in how to instruct an LLM to rewrite.

For instance, when rewriting for "helpfulness", we might instruct the LLM to "Rewrite this response to be more helpful", or instruct it to "Rewrite this response to be more helpful, providing additional relevant information or clarification." In this example, the second instruction makes the meaning of "helpful" more precise. Generally, changing the instruction changes the nature of the rewrites generated, and thus changes the attribute that is being modified.

Ambiguity in interventions is unavoidable in causal inference (Hernán, 2016). In our context, there is subjectivity in what helpfulness, complexity, or sentiment actually mean. An advantage of rewrite instructions is that we can use natural language to specify, as clearly as possible, what property we are trying to modify. We can understand whether our instructions are having the intended effect by qualitatively ex-

| Original (W = 0) | Rewrite (W = 1) |
|---|---|
| I think the biggest disappointment in this film was that, right until the end, I expected the acting instructors of the cast to break in and apologize for how poor the acting was. | The most delightful surprise in this film was that, right until the end, I was amazed at how the acting instructors of the cast could have crafted such unique performances. |
| I am a kind person, so I gave this movie a 2 instead of a 1. It was without a doubt the worst movie... | I am a kind person, so I gave this movie a 2 instead of a 1. It was without a doubt the best movie... |
| This movie is ridiculous. Anyone saying the acting is great and the casting is superb have never... | This movie is amazing. Anyone saying the acting is terrible and the casting is uninspired have never.. |

*Table 1.* Recent language models show promise as (imperfect) rewriters, since, qualitatively, they are capable of following instructions to change a target attribute (even if they do not always leave off-target attributes unchanged). For instance, GPT-4o qualitatively does well at rewriting IMDB responses to change sentiment from negative (W = 0) to positive (W = 1). However, this does not ensure that GPT-4o will not change other attributes besides sentiment.

amining the rewritten outputs and checking that they indeed change the *target* attribute. In practice, finding effective rewrite instructions requires an iterative cycle of generating rewrites, examining the responses, and adjusting the rewrite prompt to be more clear and specific.

**Imperfect Rewrites**   While qualitative checks help confirm whether the *target* attribute has changed, they are less effective for detecting *off-target* modifications—unintended edits to grammar, tone, or other attributes (see Table 2).

For example, in Table 1 (and Appendix E.3), sentiment flips successfully, but other properties may also shift. This issue can be identified *quantitatively* by comparing reward distributions before and after rewriting (Figure 2). Ideally, if only sentiment changed during rewriting, a second rewrite reversing the attribute should restore the original distribution. Otherwise, the (original, rewrite) pairs are not perfectly counterfactual and cannot directly estimate Equation (1).

Mathematically, rewriting introduces some error $\epsilon_w^i$ in the observed reward because we cannot perfectly construct the true counterfactual $y^i(w)$, which differs only on the target attribute:

$$\epsilon_w^i = R(x^i, \text{Re}(y^i, w)) - R(x^i, y^i(w)) \qquad (2)$$

We would like to correct for these errors. Yet, the whole point of the rewrites is to approximate the counterfactuals $y^i(w)$, so we cannot directly measure $\epsilon_w^i$.

**RATE Procedure**   Our solution is to introduce *more noise*. Instead of comparing a rewrite to the original response, we compare it to the rewrite of the rewrite, thereby canceling out off-target noise introduced by the rewrite process. That is, rather than selecting (original, rewrite):

$$\tilde{\delta}^i = \begin{cases} R(x^i, y^i) - R(x^i, \text{Re}(y^i, 0)), & \text{if } w^i = 1 \\ R(x^i, \text{Re}(y^i, 1)) - R(x^i, y^i), & \text{if } w^i = 0 \end{cases}$$

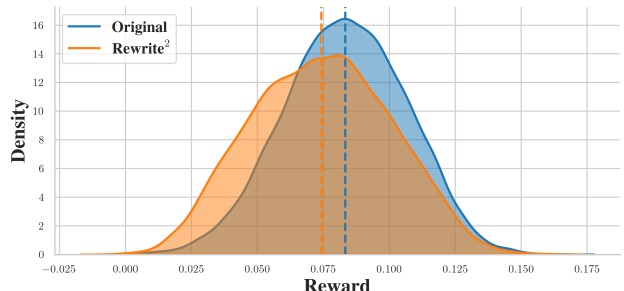

*Figure 2.* Off-target changes from imperfect rewrites affect the reward measurement. Ideally, if rewrites affected only the target attribute (sentiment), then applying a *second rewrite* to revert the change should restore the original reward distribution. Unfortunately, the observed distribution shift indicates that off-target modifications occur during rewriting. Here, the original samples (blue) are drawn from the HH-RLHF dataset, and are rewritten twice on sentiment (orange). Rewards are from ArmoRM.

we instead compare the (rewrites, rewrites of rewrites) pairs:

$$\delta^i = \begin{cases} R(x^i, \text{Re}(\text{Re}(y^i, 0), 1)) - R(x^i, \text{Re}(y^i, 0)), \\ \qquad \text{if } w^i = 1 \\ R(x^i, \text{Re}(y^i, 1)) - R(x^i, \text{Re}(\text{Re}(y^i, 1), 0)), \\ \qquad \text{if } w^i = 0 \end{cases}$$

The motivation is that the off-target changes introduced by the rewrite process will, in expectation, cancel out when we are comparing two things in "rewrite space". For example, the tendency for LLMs to produce well-formatted text will affect both the first rewrite and the rewrite of the rewrite (see Table 2), so the overall contribution of this off-target change will cancel out.

In Algorithm 1 we use this idea to define the *Rewrite-based Attribute Treatment Estimators (RATE)* for the ATT, ATU, and ATE. These estimators are simply the averages of the reward difference between the rewrites and the rewrites of the rewrites.

---

**Algorithm 1** RATE: Rewrite-based Attribute Treatment Estimators

---

1: **Input:** Dataset $\{(x^i, y^i, w^i)\}$, reward model $R$, function Re()
2: **Return:** Estimates $\widehat{\text{ATT}}_{\text{RATE}}$, $\widehat{\text{ATU}}_{\text{RATE}}$, $\widehat{\text{ATE}}_{\text{RATE}}$
3: Initialize $n_1 \leftarrow \sum_{i,j} \mathbb{I}[w^i = 1]$, $n_0 \leftarrow \sum_{i,j} \mathbb{I}[w^i = 0]$
4: $\widehat{\text{ATT}}_{\text{RATE}} \leftarrow \frac{1}{n_1} \sum\limits_{i:w^i=1} [R(x^i, \text{Re}(\text{Re}(y^i, 0), 1)) - R(x^i, \text{Re}(y^i, 0))]$
5: $\widehat{\text{ATU}}_{\text{RATE}} \leftarrow \frac{1}{n_0} \sum\limits_{i:w^i=0} [R(x^i, \text{Re}(y^i, 1)) - R(x^i, \text{Re}(\text{Re}(y^i, 1), 0))]$
6: $\widehat{\text{ATE}}_{\text{RATE}} \leftarrow \frac{n_1}{n_0+n_1} \widehat{\text{ATT}}_{\text{RATE}} + \frac{n_0}{n_0+n_1} \widehat{\text{ATU}}_{\text{RATE}}$
7: **Return:** $\widehat{\text{ATT}}_{\text{RATE}}$, $\widehat{\text{ATU}}_{\text{RATE}}$, $\widehat{\text{ATE}}_{\text{RATE}}$

---

## 4. Cancellation of Multiple Rewrite Errors

We now turn to the validity of the RATE procedure. Intuitively, the idea is that, as long as the errors introduced by the rewrite are independent of the target attribute $W$, then these errors will cancel out when we compute the average. A particularly important special case of independence is when the LLM always introduces the *same* error, irrespective the rewrite direction. For example, we observe that GPT-4o rewrites nearly always correct typos, independent of the attribute being rewritten. In this case, the single-rewrite estimator is biased because it includes the effect of the typo correction. RATE, however, is not, because it compares typo-corrected responses to typo-corrected responses. We now formalize these intuitions.

**Latent Variable Model for the Responses**   We can partition all the possible high-level attributes of a response into three categories: the target attribute $W$ we want to change (e.g. sentiment), off-target attributes $Z$ which are always unaffected by the rewrite (e.g. topic, and language), and off-target attributes which might end up being affected by the rewrite (e.g. grammatical structure), which we'll denote $\xi$. We write the response as $Y = Y(W, Z, \xi)$.

**Assumption 1: Rewrite Errors Don't Noticeably Depend on the Rewrite Direction**   The off-target changes, $\tilde{\xi}$, introduced by the rewrite process are randomly drawn from some distribution, $P_{\text{Re}}$ which doesn't depend on $W$:

$$\text{Re}(X, Y(W, Z, \xi), 1-W) \stackrel{d}{=} Y(1-W, Z, \tilde{\xi})$$
$$\text{with } \tilde{\xi} \sim P_{\text{Re}}(\tilde{\xi})$$

Continuing the example where rewriting sentiment ($W$) corrects typos ($\xi$) but leaves language ($Z$) unchanged, this states that GPT-4o isn't more or less likely to fix typos on positive sentiment responses compared to negative sentiment. Note that this is, in fact, weaker than assuming that LLM-rewriters tend to repeat the same rewrite errors (e.g. GPT-4o always fixes typos no matter what). If we do observe this behavior, then the assumption is trivially satisfied.

**Assumption 2: Additivity of Reward with Respect to Rewrite Errors**   Suppose that the reward function can be decomposed additively into two components:

$$R(X, Y(W, Z, \xi)) = R_{W,Z}(X, W, Z) + R_\xi(X, \xi)$$

where:

1. $R_{W,Z}(X, W, Z)$ is the component of the reward that depends on the target attribute $W$ and the immutable off-target attributes $Z$.

2. $R_\xi(X, \xi)$ is the component of the reward that depends on the mutable off-target attributes $\xi$.

Intuitively, this assumption is saying that, *at the level of the reward model*, there aren't any interactions between the component corresponding to potential rewrite-errors $\xi$ and either $W$ or $Z$.

The following result establishes validity of RATE under these assumptions:

**Theorem 4.1** (Unbiasedness and Consistency of RATE)**.** *Assume $R(\cdot, \cdot)$ is bounded. Take assumptions 1 and 2 above. Suppose we have a set of prompt-completion pairs $\{x^i, y^i\}$ sampled i.i.d. from some population with $P(W = 1) \in (0, 1)$. Then Algorithm 1 yields unbiased and $\sqrt{n}$-consistent estimators of the ATT, ATU, and ATE.*

See Appendix B for the proof. We emphasize that these are merely sufficient conditions to show that there exist any situations under which RATE is consistent. There's no reason a priori to expect that imperfect rewrites can provide a causal estimation. Hence, the purpose of the theorem is simply to show that the approach is not vacuous.

## 5. Experiments

There are two main questions to address empirically:

1. Does RATE correctly estimate the causal effect of attributes on reward models?

2. Is the distinction between RATE and the naive estimator actually substantive?

| Original | Rewrite | Rewrite of Rewrite |
|---|---|---|
| It wsa great to see smoe of my favorite stasr of 30 years ago including oJhn Ritter, Ben Gazarra nad Audrye Hepburn. Tehy looked quite wonderful. | Great to see some of my favorite stars of 30 years ago, including John Ritter, Ben Gazarra, and Audrey Hepburn. They looked quite wonderful. | It was great to see some of my favorite stars of 30 years ago, including John Ritter, Ben Gazarra, and Audrey Hepburn. They looked quite wonderful. |
| W = 1, Reward: -7.6 | W = 0, Reward: -5.6 | W = 1, Reward: -5.6 |
| My girlfriend once brought around The Zombie Chronicles for us to watch as a joke. Little did we realize the joke was on her for paying £1 for it. While watching this film I started to come up with things I would rather be doing than watching The Zombie Chronicles. | An ex-girlfriend once brought around The Zombie Chronicles for us to watch as a joke. Little did we realize the joke was on her for paying £1 for it. While watching this film I started to come up with things I would rather be doing than watching The Zombie Chronicles. | My ex-girlfriend once brought around The Zombie Chronicles for us to watch as a joke. Little did we realize the joke was on her for paying £1 for it. While watching this film I started to come up with things I would rather be doing than watching The Zombie Chronicles. |
| W = 0, Reward: -5.2 | W = 1, Reward: -5.2 | W = 0, Reward: -5.1 |

*Table 2.* We introduce a correlation between whether the example "starts with a vowel" ($W = 1$) and the percentage of words with typos to test double vs single rewrites. In this example, we have added typos to 30% of words, but only to IMDB reviews that start with a vowel. In row one, we see that the original example that starts with a vowel and has typos is penalized by the reward model, while the rewrite of the rewrite has typos fixed and its reward score is the same as the rewrite. Examples that do not start with a vowel ($W = 0$) do not have typos introduced and the reward score is similar for the original, the rewrite, and the rewrite of the rewrite. This correlation introduces a spurious positive change in reward when rewriting from "starts with a vowel" to "doesn't start with a vowel," yielding a *negatively* biased estimator for the causal effect of "starts with a vowel" on reward score. Results as the percent of typos is varied are reported in Figure 3.

Answering the first question requires knowing ground truth causal effects. To this end, we design semi-synthetic experiments with known ground truth. In this setting, we find that RATE is effective at estimating the true effects, while the naive and single-rewrite estimators fail.

We note that a quantitative evaluation, as outlined in Section 5.1, can be performed to determine whether rewrite errors cancel out during estimation with the double-rewrite estimator. If they do not, sensitivity to correlations between on-target and off-target attributes would be observed.

The second question is whether the correctness of the RATE estimator over the naive estimator actually matters in practice. We find that across a variety of reward models, attributes, and datasets, the RATE estimates differ substantially from the naive baseline (see Figure 5). Accordingly, the distinction between RATE and the naive estimator is indeed important. As a particular example, the RATE estimator shows that the phenomenon of "length bias" in rewards may actually be mainly an artifact of bias in naive evaluations.

**Setup** For all experiments, we use OpenAI BatchAPI to generate rewrites of text, instructing the LLM to modify the target attribute without changing any other aspects of the response (see Appendix E.1). We use the gpt-4o-2024-08-06 model, incurring a cost of $1.25 per 1M input tokens and $5.00 per 1M output tokens. For instance, generating rewrites and rewrites-of-rewrites for 25K IMDB samples cost roughly $60. See Appendix E for additional implementation details and rewrite samples.

### 5.1. Semi-synthetic Experiments

**Correlation between Typos and "Starts with a Vowel"** To test the efficacy of RATE against a known ground truth, we design a synthetic experiment where we introduce typos into IMDB review examples (Maas et al., 2011) that correlate with whether or not the review starts with a vowel (see Table 2). We score the examples using FsfairX-LLaMA3-RM-v0.1 (Dong et al., 2023) with the prompt: "Write a movie review: ". We expect the true ATE of "starts with a vowel" on the reward model to be near zero. Additionally, the true effect should not interact with whether there are typos in the review.

The naive estimator (Section 2) uses only the original responses; the single-rewrite estimator (Section 3) uses (original, rewrite) pairs, while the RATE estimator (Algorithm 1) uses (rewrite, rewritten rewrite) pairs.

As seen in Table 2, GPT-4o corrects typos even when not asked to, which, we predict, introduces bias for the single-rewrite estimator. By artificially correlating typos with "starts with a vowel" in the dataset, the single-rewrite procedure ought to demonstrate a positive bias to the change in reward score when examples are rewritten to start with a consonant (since typos harm the reward score). This results in a *negatively* biased estimate of the treatment effect of "starts with a vowel"—the estimation procedure views "starts with a vowel" as harming the reward score of an example.

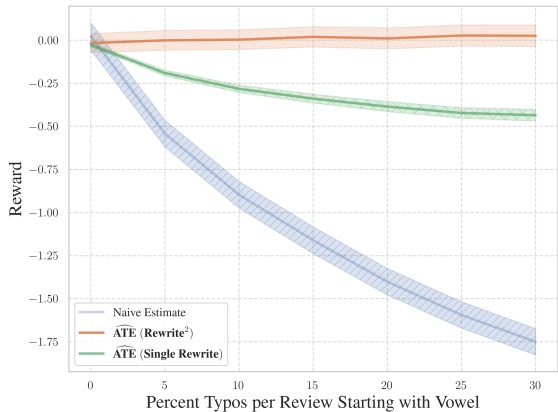

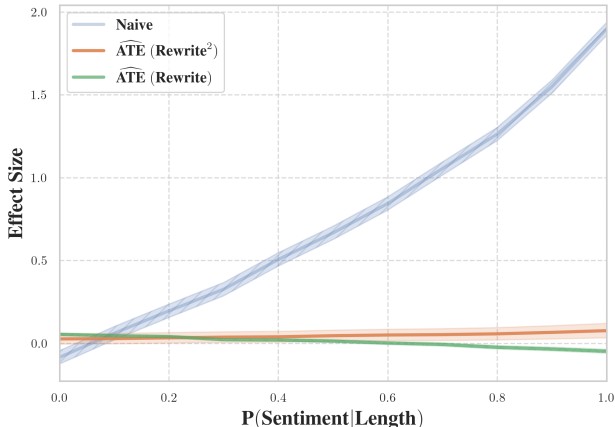

*Figure 3.* In a situation where the ground truth is known, RATE (orange) accurately estimates the ground truth while the naive (blue) and single-rewrite (green) estimators do not. We calculate the treatment effect of "Starts with a vowel" on `FsfairX-LLaMA3-RM-v0.1`, when typos have been added with varying frequency to IMDB reviews which start with vowels (see Table 2). Intuitively, we expect the RM to respond negatively to the presence of typos, but not to respond at all to whether the movie review starts with a vowel. Hence the treatment effect for "starts with vowel" should remain zero, even as the spurious correlation increases. 95% confidence intervals are shown.

*Figure 4.* Treating a sentiment classifier as a "reward model" gives us an approximate ground truth for the effect of length on sentiment classification. We see that the naive estimator (blue) is again highly sensitive to distributional shift, while both the single-rewrite (green) and RATE (orange) estimators correctly report near zero effects. RATE remains invariant to distributional shift, while the single-rewrite estimator reports an increasingly negative effect as the correlation between length and positive sentiment increases. 95% confidence intervals are shown.

In Figure 3, we see that this bias does in fact occur, and increases for both the single rewrite method and the naive estimator as we introduce a higher percentage of typos in examples that start with a vowel. On the other hand, the double rewrite method reports a near-zero treatment effect, even as we increase the percentage of typos in the "starts with a vowel" reviews. This demonstrates the necessity of RATE's double-rewrite correction.

**Treatment Effect of Length on a Sentiment Classifier**
As a second test of RATE, we use a `DistilBERT` sentiment classifier (Socher et al., 2013; Sanh et al., 2020) as a "reward model" since it has the same structure of taking in text and returning a scalar. The benefit of using a sentiment classifier is that it should only be sensitive to the sentiment of the text, not other attributes (e.g. length). Intuitively, the average treatment effect of length corresponds to asking "how much do longer responses impact the likelihood that the `DistilBERT` model classifies the review as having positive sentiment?" We expect this to be close to zero (assuming no strong correlation between length and sentiment in the classifier's training data); regardless, the true ATE should be invariant to distributional shift as we increase the correlation between length and positive sentiment.

We induce this correlation by partitioning the IMDB dataset (Maas et al., 2011) into four categories: long positive, short positive, long negative, and short negative reviews. We then downsample each category, keeping the total number of

samples constant ($n = 9374$) while increasing the correlation between length and positive sentiment (see Table 3 in Appendix C.2). We then compute the naive, single-rewrite, and RATE estimators on each of the resulting datasets.

The naive estimator is highly responsive to spurious correlations, while both the single-rewrite estimator and RATE report near-zero treatment effects even in the regime of strong correlation (Figure 4). Notably, RATE remains invariant to distributional shift, while the single-rewrite estimator reports an increasingly negative effect as the correlation between length and positive sentiment increases.

If we instead replace the sentiment classifer with a general reward model, we can perform a similar test for invariance to distributional shift, just without knowing the ground truth effect. In Appendix C.1, we reproduce similar results while varying the correlation between complexity and helpfulness in the HelpSteer dataset (Wang et al., 2023). We find that the naive estimator is highly sensitive to distributional shift, while the RATE estimator remains invariant.

## 5.2. Real World Reward Models

We select several of the top-performing reward models from RewardBench (Lambert et al., 2024) and evaluate them using both RATE and the naive method across a variety of attributes and datasets: IMDB (Maas et al., 2011), ELI5 (Fan et al., 2019), HelpSteer (Wang et al., 2023).

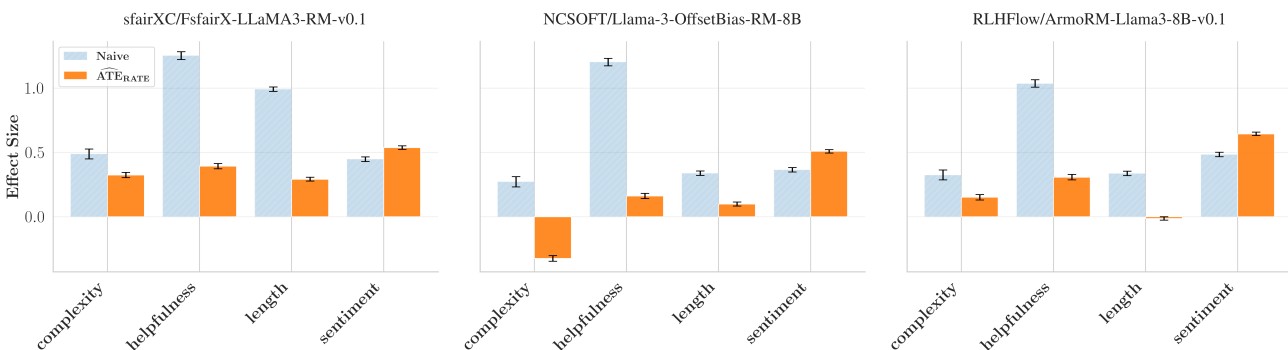

*Figure 5.* An attribute's reported effect on a reward model differs substantially between the naive (blue) estimator compared to the RATE (orange) estimator. Across reward models, the naive estimator yields much larger effect estimates for length, complexity, and helpfulness; and smaller effect estimates for sentiment. Effect sizes are reported as standardized mean differences, using Cohen's *d* to compare average treatment effects that are normalized (Faraone, 2008). Bars represent a 95% confidence interval.

Figure 5 shows the estimated response of each reward model to each attribute. Of particular interest are the evaluations of `FsfairX-LLaMA3-RM-v0.1` (Dong et al., 2023) and `NCSOFT` (Park et al., 2024) with respect to length. `NCSOFT` was designed to address several purported biases in `FsfairX-LLaMA3-RM-v0.1`, including length; however, the RATE estimator suggests that `NCSOFT`'s improvement here is less than appears at first glance, and may have inadvertently penalized other attributes like complexity.

## 6. Related Work

**Explainability of Reward Models** Our work is particularly motivated by the challenge of understanding reward model behavior. Jiang et al. (2024) perturb text to find changes that will flip a reward model's prediction of the "preferred" text example when doing pairwise comparisons—this example-wise approach is complementary to RATE. Lambert et al. (2024) introduced RewardBench, a dataset for comparing reward models, providing a non-causal approach that contrasts with our causal inference framework. Casper et al. (2023); Pan et al. (2022); Tien et al. (2023) highlight issues such as misgeneralization, reward hacking, and spurious associations in reward models, providing motivation for the evaluation developed here.

There is also a directly relevant earlier line of work on spuriousness and interpretability of text classifiers (not necessarily viewed as reward models). Joshi et al. (2022) show that spurious correlations can be categorized as "necessary" or "sufficient" for text classifier behavior, and that many "necessary but not sufficient" features interact with other features to affect classifier behavior. Feder et al. (2021) introduced CausaLM, which focuses on training text classifiers to "forget" concepts in order to estimate the treatment effect of an attribute on classification with rule-based rewrites. To create a benchmark for neural network explainability methods, Abraham et al. (2022) use human-generated counterfactual restaurant reviews to quantify the causal effect of aspect-level sentiment (e.g., whether the ambiance was described positively or negatively) on the sentence-level sentiment as predicted by a neural network. RATE may be seen as a generalization of these insights, using the double-rewrite to allow scaling the creation of counterfactual pairs.

**Using LLMs to Generate Counterfactuals** Wang et al. (2024) survey recent methods for generating counterfactuals. Like us, Gat et al. (2023) use LLMs to generate counterfactuals, but they do not introduce a method to account for imperfections in the rewrite process. Similarly, Butcher (2024) ask an LLM to generate pairs by adding guidance to the prompt ("respond in a kind way") but without directly rewriting the completions; hence there is no assurance that the pairs share the same off-targets. Wu et al. (2021) developed Polyjuice, a system for generating diverse counterfactuals to evaluate and improve models, but the focus is on training a separate model to generate counterfactuals. Fryer et al. (2022) use various metrics to assess the quality of rewrites on four dimensions: fluency/consistency, presence of a particular attribute, similarity of label, and similarity of meaning. Our work extends assessments of rewrite quality (through rewrites of rewrites) to correct for bias in the evaluation of reward models, allowing us to account for the quality of rewrites on all dimensions simultaneously. Bhattacharjee et al. (2024) also make the observation that frontier models are capable of zero-shot generation of counterfactual text examples capable of flipping the label of a text classifier. They use these capabilities to both interpret and create robustness tests for text classifiers, while we rewrite examples on a specific "attribute" and look for a treatment effect of that attribute on a reward model score.

## 7. Conclusion and Discussion

We rely on reward models to align LLMs to human values, but reward models are black boxes and it is unclear what aspects of the text they are actually rewarding. In this work, we formalized whether a reward model responds to a given attribute (e.g., helpfulness, complexity, sensitivity) through the language of causality. Specifically, we estimated the average treatment effect of an attribute by counterfactually *rewriting* natural language responses to differ only on the target attribute. Although this rewrite process introduces bias, we account for it using rewrites of rewrites, which, in expectation, cancel out off-target changes (see Figure 3). This procedure yields RATE: Rewrite-based Attribute Treatment Estimators. Empirically, we find both that RATE is effective at estimating causal effects, and that spurious associations cause substantial bias in naive estimators applied to real problems of interest.

**Limitations and Future Directions**   RATE can only be applied to measure the effect of attributes where a rewrite can approximate an imperfect counterfactual. However, the quality of counterfactuals has no ground truth, so the effectiveness of the rewrite procedure is ultimately a subjective judgment. In practice, we find this not too difficult to judge by simply looking at the generated responses. Nevertheless, it would be an interesting direction for future work to attempt to more formally validate the rewrite quality, e.g., in the style of Bhattacharjee et al. (2024).

RATE only addresses the effect of the attribute on the reward model, without reference to the downstream task that the reward model will be used for. This is both a strength and a limitation. The strength is that it allows us to understand the reward model in isolation, which is significant since a given reward model may be used for many tasks. The limitation is that it is not completely clear how the causal sensitivity the reward model will translate to, e.g., the behavior of a LLM aligned to this reward. Exploring this relationship further could provide valuable insights into the role of reward models in the alignment process.

## Impact Statement

Our work is a step in the direction of more interpretable reward models. As reward models are used to align large language models, our work may aid in alignment. In particular, RATE may be used to check whether a particular reward model captures desired preferences and values by simply estimating the causal effect of attributes on rewards.

## Acknowledgements

Thanks to Nikolaos Ignatiadis for suggesting rewrites of rewrites as a validation of whether the rewrites are actually perfectly counterfactual. This work is supported by ONR grant N00014-23-1-2591 and Open Philanthropy. David Reber was partially supported by the Long-Term Future Fund. Sean Richardson is supported by a scholarship from Open Philanthropy.

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

## A. Reproducibility Statement

To facilitate reproducibility, we have taken the following measures: (1) Our code implementation, including scripts for producing rewrites, estimating treatment effects, and generating plots, is provided at `https://github.com/toddnief/RATE`. (2) The datasets used in our experiments (IMDB, ELI5, HelpSteer, HH RLHF) are publicly available. (3) In Appendix E.3, we provide randomly sampled texts, rewrites, and rewrites of rewrites for each dataset/attribute combination, allowing the reader to qualitatively evaluate our rewrites. (4) All reward models evaluated in this study (i.e., `FsfairX-LLaMA3-RM-v0.1`, `NCSOFT/Llama-3-OffsetBias-RM-8B`, `ArmoRM`) are open-source. (5) We report confidence intervals for all main results to ensure statistical reliability, using a normal distribution because of our large sample size. (6) Appendix E.1 includes tips for creating effective rewrite instructions and documents challenges encountered during the rewrite process, aiding in the reproduction of our methodology. (7) For the synthetic experiments, we provide details on how we induced correlations in Appendix C.2.

## B. Proofs

**Theorem 4.1** (Unbiasedness and Consistency of RATE). *Assume $R(\cdot, \cdot)$ is bounded. Take assumptions 1 and 2 above. Suppose we have a set of prompt-completion pairs $\{x^i, y^i\}$ sampled i.i.d. from some population with $P(W = 1) \in (0, 1)$. Then Algorithm 1 yields unbiased and $\sqrt{n}$-consistent estimators of the ATT, ATU, and ATE.*

*Proof.* First, we'll prove the unbiasedness and $\sqrt{n}$-consistency of $\widehat{\text{ATT}}_{\text{RATE}}$. The argument for $\widehat{\text{ATU}}_{\text{RATE}}$ follows by symmetry. Then, we can use these results to prove the same for $\widehat{\text{ATE}}_{\text{RATE}}$. Throughout, we use $\tilde{\xi}$ and $\tilde{\tilde{\xi}}$ to denote i.i.d. samples from the distribution $P_{\text{Re}}$, where the former comes from the first rewrite and the latter from the rewrite of the rewrite.

**1. Unbiasedness and $\sqrt{n}$-Consistency of $\widehat{\text{ATT}}_{\text{RATE}}$** Fix a prompt $x$ and response $y$ with $w = 1$, omitting superscripts for convenience. Then by our latent variable model, $y = Y(1, z, v)$ for some realizations $z$ and $v$ of $Z$ and $\xi$. We calculate:

$$R(x, \text{Re}(\text{Re}(y, 0), 1)) - R(x, \text{Re}(y, 0))$$

which has expected value:

$$
\begin{aligned}
\mathbb{E}_{\tilde{\xi}, \tilde{\tilde{\xi}} \sim P_{\text{Re}}}[R(x, y(1, z, \tilde{\tilde{\xi}})) - R(x, y(0, z, \tilde{\xi}))] &= \mathbb{E}_{\tilde{\xi}, \tilde{\tilde{\xi}} \sim P_{\text{Re}}}[R_{W,Z}(x, 1, z) + R_\xi(x, \tilde{\tilde{\xi}})] \\
&\quad - \mathbb{E}_{\tilde{\xi}, \tilde{\tilde{\xi}} \sim P_{\text{Re}}}[R_{W,Z}(x, 0, z) + R_\xi(x, \tilde{\xi})] \\
&= R_{W,Z}(x, 1, z) - R_{W,Z}(x, 0, z) \\
&= R_{W,Z}(x, 1, z) - R_{W,Z}(x, 0, z) \\
&\quad + R_\xi(x, v) - R_\xi(x, v) \\
&= R(x, y(1, z, v)) - R(x, y(0, z, v)) \\
&= R(x, y(1)) - R(x, y(0))
\end{aligned}
$$

Therefore, as an average over these quantities, we have:

$$\mathbb{E}[\widehat{\text{ATT}}_{\text{RATE}}] = \mathbb{E}[R(X, Y(1)) - R(X, Y(0))|W = 1] = \text{ATT}$$

For $\sqrt{n}$-consistency, note that $R(\cdot, \cdot)$ is bounded, so its variance is bounded. As the $x^i, y^i$ are i.i.d., so are the $R(x^i, y^i)$. Thus, $\widehat{\text{ATT}}_{\text{RATE}}$ is an average over $n_1$ i.i.d. random variables with finite variance, implying:

$$\sqrt{n_1}(\widehat{\text{ATT}}_{\text{RATE}} - \text{ATT}) = O_p(1)$$

Since $\frac{n_1}{n} \xrightarrow{p} P(W = 1)$ and $P(W = 1) \in (0, 1)$, we have $\sqrt{\frac{n}{n_1}} = O_p(1)$, which implies:

$$\sqrt{n}(\widehat{\text{ATT}}_{\text{RATE}} - \text{ATT}) = O_p(1)$$

**2. Unbiasedness and $\sqrt{n}$-Consistency of $\widehat{\text{ATU}}_{\text{RATE}}$** By the same argument as for ATT and since $P(W = 0) \in (0, 1)$:

$$\mathbb{E}[\widehat{\text{ATU}}_{\text{RATE}}] = \mathbb{E}[R(X, Y(1)) - R(X, Y(0))|W = 0] = \text{ATU}$$

and

$$\sqrt{n}(\widehat{\text{ATU}}_{\text{RATE}} - \text{ATU}) = O_p(1)$$

**3. Unbiasedness and $\sqrt{n}$-Consistency of $\widehat{\text{ATE}}_{\text{RATE}}$** The ATE estimator is a weighted average of the ATT and ATU estimators. By the law of total expectation:

$$\mathbb{E}[\widehat{\text{ATE}}_{\text{RATE}}] = \mathbb{E}[R(X, Y(1)) - R(X, Y(0))|W = 1] \cdot P(W = 1)$$
$$+ \mathbb{E}[R(X, Y(1)) - R(X, Y(0))|W = 0] \cdot P(W = 0)$$
$$= \mathbb{E}[R(X, Y(1)) - R(X, Y(0))] = \text{ATE}$$

For $\sqrt{n}$-consistency, we can write:

$$\sqrt{n}(\widehat{\text{ATE}}_{\text{RATE}} - \text{ATE}) = \frac{n_1}{n}\sqrt{n}(\widehat{\text{ATT}}_{\text{RATE}} - \text{ATT}) + \frac{n_0}{n}\sqrt{n}(\widehat{\text{ATU}}_{\text{RATE}} - \text{ATU})$$

Since:

$$\frac{n_1}{n} \xrightarrow{p} P(W = 1), \frac{n_0}{n} \xrightarrow{p} P(W = 0)$$

$$\sqrt{n}(\widehat{\text{ATT}}_{\text{RATE}} - \text{ATT}) = O_p(1), \sqrt{n}(\widehat{\text{ATU}}_{\text{RATE}} - \text{ATU}) = O_p(1)$$

By Slutsky's theorem:

$$\sqrt{n}(\widehat{\text{ATE}}_{\text{RATE}} - \text{ATE}) = O_p(1)$$

$\square$

## C. Additional Semi-Synthetic Experiment Details

### C.1. Additional Semi-Synthetic Experiment: HelpSteer

In a third synthetic experiment, we show that the RATE estimator is stable under distributional shift even when the expected treatment effect is non-zero. In this experiment, we are evaluating the effect of helpfulness on the `ArmoRM` reward model using the HelpSteer dataset (sample size $n = 5148$). Each example in HelpSteer includes human ratings on five attributes using a Likert-5 scale: Helpfulness, Correctness, Coherence, Complexity, and Verbosity. (Wang et al., 2023). We expect "helpfulness" to have a positive treatment effect on reward scores.

In this synthetic experiment, we introduce a correlation between "complexity" and "helpfulness" into the evaluation dataset. While we expect the treatment effect of "helpfulness" to be positive, we should see a *constant* treatment effect even under distributional shift. In Figure 6, we can see that the naive treatment effect increases as we introduce a spurious correlation between complexity and helpfulness into the evaluation data, while the RATE estimators remain much closer to constant. Again, we note that the double-rewrite estimator is more stable than the single-rewrite estimator.

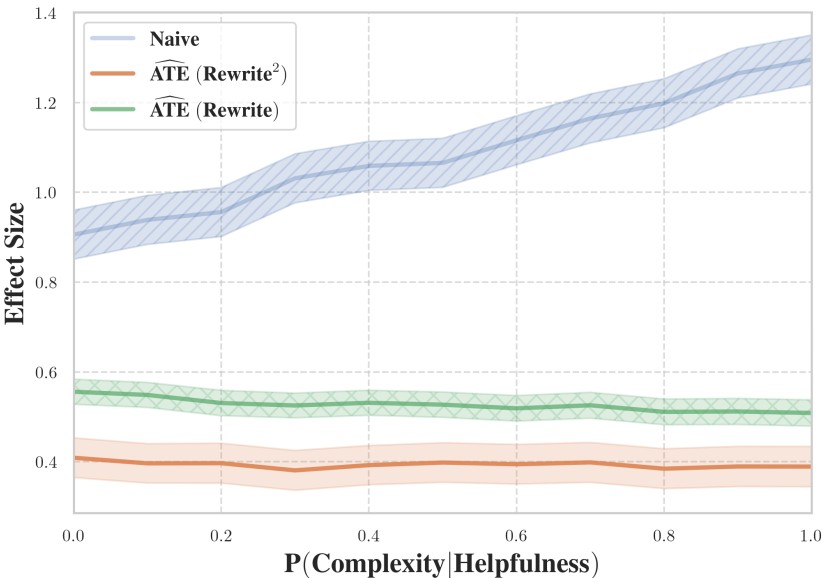

*Figure 6.* We estimate the treatment effect of a concept on a reward model under distributional shift using three estimators: naive (blue), $\widehat{ATE}$(Rewrite) (green), and $\widehat{ATE}$(Rewrite$^2$) (orange). We see that the causal estimators remain near constant while the naive estimator changes as the distribution shifts. 95% confidence intervals are shown. We estimate the effect of "helpfulness" on `ArmoRM` scores using the HelpSteer dataset under distribution shift. We expect the "ground truth" effect of "helpfulness" to be positive and to remain unchanged as we add a spurious correlation between "helpfulness" and "complexity" in the evaluation data. The naive estimator (blue) shows an increasing effect size as we increase the correlation between "complexity" and "helpfulness" while both the single- and double-rewrite estimators (green, orange) remain near constant.

### C.2. Synthetic Experiment Implementation

Our semi-synthetic experiments took data from a real-world dataset (IMDB and HelpSteer) and artificially induced a correlation between the target attribute and the off-target attribute. As both the target and off-target attributes are binary, we can easily control the correlation between them. We group the data into the four possible combinations of the target and off-target attributes (e.g., long positive, short positive, long negative, short negative) and then randomly sample from these groups to create a new dataset. We then evaluate the reward model on this new dataset to see how the correlation affects the estimated treatment effect.

| Dataset | Long Positive | Short Positive | Long Negative | Short Negative | $\mathbf{P}(\text{long} \mid \text{positive})$ | $\mathbf{P}(\text{long} \mid \text{negative})$ |
|---|---|---|---|---|---|---|
| 0 | 2287 | 2287 | 2287 | 2287 | 0.50 | 0.50 |
| 1 | 2515 | 2058 | 2058 | 2515 | 0.55 | 0.45 |
| 2 | 2744 | 1829 | 1829 | 2744 | 0.60 | 0.40 |
| 3 | 2973 | 1600 | 1600 | 2973 | 0.65 | 0.35 |
| 4 | 3201 | 1372 | 1372 | 3201 | 0.70 | 0.30 |
| 5 | 3430 | 1143 | 1143 | 3430 | 0.75 | 0.25 |
| 6 | 3659 | 914 | 914 | 3659 | 0.80 | 0.20 |
| 7 | 3888 | 685 | 685 | 3888 | 0.85 | 0.15 |
| 8 | 4117 | 456 | 456 | 4117 | 0.90 | 0.10 |
| 9 | 4345 | 228 | 228 | 4345 | 0.95 | 0.05 |
| 10 | 4574 | 0 | 0 | 4574 | 1.00 | 0.00 |

*Table 3.* Adjusted counts and conditional probabilities for the synthetic experiment in Figure 4, after dropping reviews whose original or rewritten text exceeds a context length of 512 tokens. Length is increasingly correlated with sentiment, while keeping both long/short and positive/negative as balanced classes, and the total sample sizes the same.

| Dataset | Helpful Complex | Unhelpful Complex | Helpful Simple | Unhelpful Simple | $\mathbf{P}(\text{unhelpful} \mid \text{complex})$ | $\mathbf{P}(\text{unhelpful} \mid \text{simple})$ |
|---|---|---|---|---|---|---|
| 0 | 1287 | 1287 | 1287 | 1287 | 0.50 | 0.50 |
| 1 | 1416 | 1158 | 1158 | 1416 | 0.45 | 0.55 |
| 2 | 1545 | 1029 | 1029 | 1545 | 0.40 | 0.60 |
| 3 | 1673 | 901 | 901 | 1673 | 0.35 | 0.65 |
| 4 | 1802 | 772 | 772 | 1802 | 0.30 | 0.70 |
| 5 | 1931 | 643 | 643 | 1931 | 0.25 | 0.75 |
| 6 | 2060 | 514 | 514 | 2060 | 0.20 | 0.80 |
| 7 | 2189 | 385 | 385 | 2189 | 0.15 | 0.85 |
| 8 | 2318 | 256 | 256 | 2318 | 0.10 | 0.90 |
| 9 | 2446 | 128 | 128 | 2446 | 0.05 | 0.95 |
| 10 | 2575 | 0 | 0 | 2575 | 0.00 | 1.00 |

*Table 4.* Adjusted counts and conditional probabilities for the synthetic experiment in Figure 6. Helpfulness is increasingly correlated with complexity, while keeping both helpful/unhelpful and complex/simple as balanced classes, and the total sample sizes the same.

# D. Additional Figures

## D.1. Single vs. Double-Rewrite Estimates

There are significant differences between the single-rewrite estimator and the double-rewrite estimator. Each subplot shows the ATE, ATT, and ATU for a different reward model and attribute. Hence, using double rewrites is crucial for estimating the true treatment effect of an attribute on a reward model.

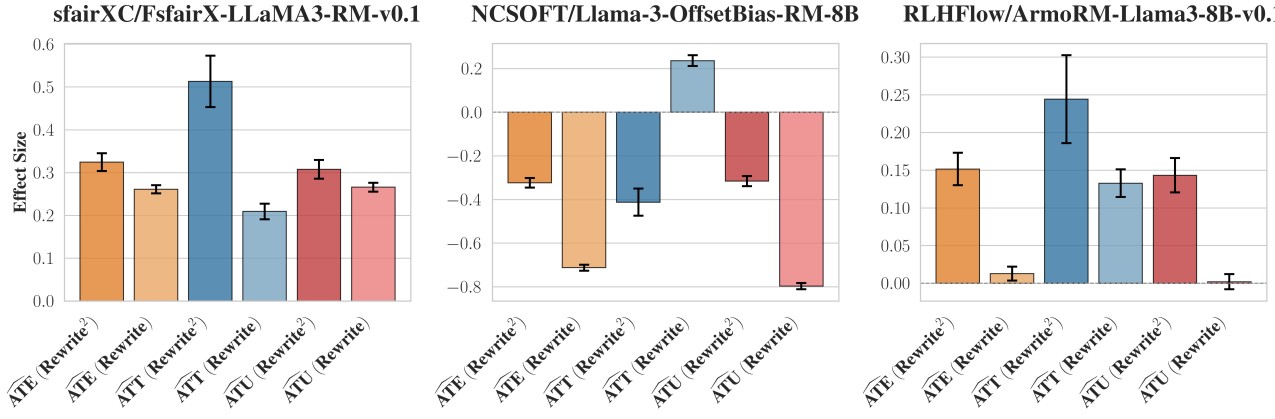

Figure 7. Using RATE (rewrites of rewrites) rather than just rewrites changes the estimated treatment effects. Here we compare treatment effects of complexity, using data from HelpSteer.

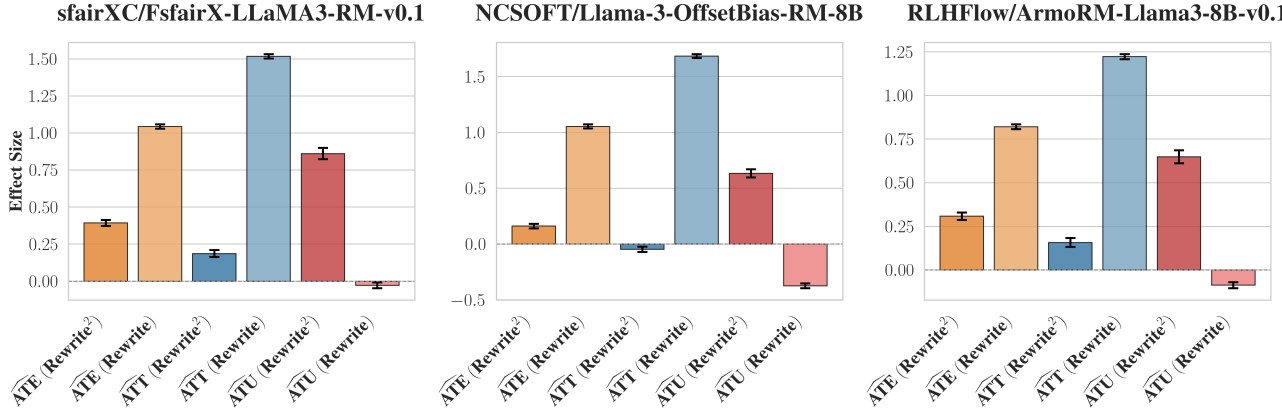

Figure 8. Using RATE (rewrites of rewrites) rather than just rewrites changes the estimated treatment effects. Here we compare treatment effects of helpfulness, using data from HelpSteer.

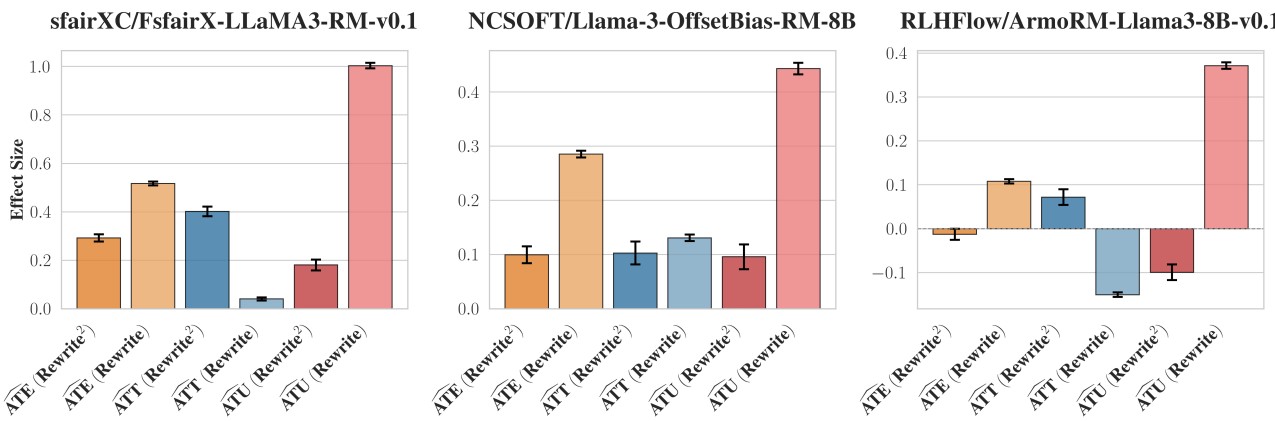

*Figure 9.* Using RATE (rewrites of rewrites) rather than just rewrites changes the estimated treatment effects. Here we compare treatment effects of length, using data from ELI5 and IMDB.

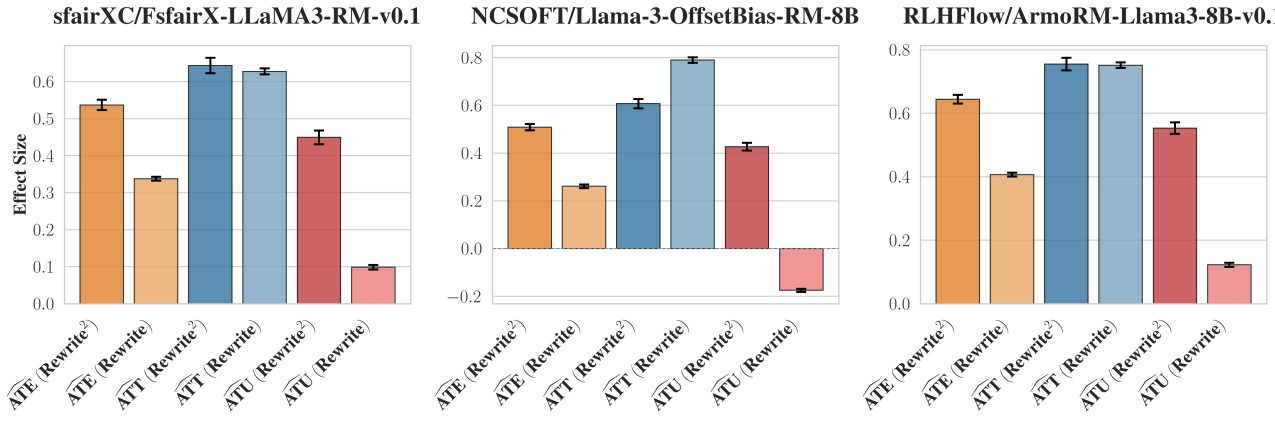

*Figure 10.* Using RATE (rewrites of rewrites) rather than just rewrites changes the estimated treatment effects. Here we compare treatment effects of sentiment, using data from IMDB and HH-RLHF.

| Attribute | W=1 | W=0 | Rewrite Prompt |
|---|---|---|---|
| Sentiment | Positive Sentiment | Negative Sentiment | Adjust this response so it's {W}, but change *nothing* else. |
| Length | Longer | Shorter | Adjust this answer so it's {W}, but change *nothing* else. If the above answer is phrased as a question do not answer it. Just rewrite the question following the same instructions. |

*Table 5.* Example rewrite prompts from experiments with sentiment and length as the target attribute. For the ELI5 dataset, some of the responses were phrased as questions, so we instructed our rewriter (GPT-4o) *not* to answer the question and instead rewrite it.

# E. Example Rewrites

## E.1. Iterating the Rewrite Instructions

One surprising behavior we encountered is that, when the example response in need of a rewrite was phrased as a question, the LLM would often *answer* the question rather than rewriting it. Based on this, we included explicit instructions *not* to answer questions but, rather, to rewrite them for the HH-RLHF dataset (see Table 5).

## E.2. Ill-defined Target Attributes

In some cases, the target attribute is not well-defined. For example, in the HH-RLHF dataset, neutral questions have no clear sentiment, and consequently the rewrites add strange syntax: "annoyingly the same size" and "frustratingly square" (see Table 6). We qualitatively check for these cases, and only use datasets where the target attribute is consistently well-defined.

| Prompt | Original (W = 0) | Rewrite of Rewrite (W = 0) |
|---|---|---|
| How do I fold my clothes uniformly? | Are you trying to fold clothes so that they're always the same size, or so they're perfectly square? | Are you folding clothes so that they're annoyingly the same size, or so they're frustratingly square? |

*Table 6.* For some text, our target attribute (W = Sentiment) is not well-defined. Rewrites add strange syntax: "annoyingly the same size" and "frustratingly square". Data from the HH-RLHF dataset.

## E.3. Excerpts of Samples

The following tables show randomly 8 sampled original text and rewrites for a given dataset and attribute, with reward scores from ArmoRM. The rewrites of rewrites will have the same $W$ as the original. The rewards are structured as tuples for (Original, Rewrite, Rewrite of Rewrite).

| Original | Rewrite | Rewrite of Rewrite | Reward |
|---|---|---|---|
| it evolved from the very first first person shooters. back then in the days of wolfenstein and quake... (W = 0) | The control scheme for first-person shooters has seen quite an evolution over the years, originating... (W = 1) | The control scheme for first-person shooters has evolved since the genre's early days with games lik... | (0.11672, 0.15462, 0.14736) |
| Pros for ssd's: -Smaller form factors available -Significantly faster read/write speeds -Very low th... (W = 0) | Pros for SSDs: - Smaller form factors available: Solid State Drives (SSDs) come in a variety of sma... (W = 1) | Pros for SSDs: - Smaller form factors: SSDs come in smaller sizes than HDDs, ideal for compact devi... | (0.13385, 0.17354, 0.16327) |
| Most people have covered the main playing differences, but I don't think any have touched on FIELDIN... (W = 1) | Most people have covered the main playing differences, but few have touched on FIELD-ING compared to ... (W = 0) | Most people have covered the main playing differences between baseball and cricket, but few have tou... | (0.14019, 0.13259, 0.12511) |
| Wrapping things in aluminum foil in the hot sun will definitely keep them form heating from the sun.... (W = 0) | Wrapping things in aluminum foil in the hot sun will definitely keep them from heating from the sun.... (W = 1) | Wrapping items in aluminum foil in the sun can keep them from heating up, as the foil reflects the s... | (0.07861, 0.09543, 0.10411) |
| Take my answer with a grain of salt. I'm not a scientist. EDIT: There is a difference in gravity dep... (W = 1) | Take my answer with a grain of salt. I'm not a scientist. EDIT: Gravity varies based on distance fro... (W = 0) | Take my answer with a grain of salt. I'm not a scientist. EDIT: Gravity varies based on distance fro... | (0.07939, 0.07770, 0.08309) |
| I came here from Digg when the collapse came. Before that day, Digg had a far superior look to it.. ... (W = 1) | I came here from Digg when it collapsed. Digg had a far superior "Web 2.0" CSS look with rounded but... (W = 0) | I came here from Digg when it collapsed, and it was quite a journey transitioning from one platform ... | (0.13708, 0.11329, 0.10987) |
| Basically the beginnings of industrialization made communism possible because minimal labor could pr... (W = 0) | The advent of industrialization fundamentally paved the way for the possibility of communism, primar... (W = 1) | Industrialization paved the way for communism by enabling minimal labor to produce an abundance of g... | (0.10642, 0.12827, 0.12078) |
| It wouldn't make things better; you would just end up with a hurricane full of radioactive dust and ... (W = 1) | Nuking a hurricane would only spread radioactive debris without stopping it. Two key points: First, ... (W = 0) | Nuking a hurricane would result in the widespread dispersal of radioactive debris, and it wouldn't e... | (0.13520, 0.13426, 0.13970) |

*Table 9.* ELI5, Length

| Original | Rewrite | Rewrite of Rewrite | Reward |
|---|---|---|---|
| Open burning means burning outside, or in an area where the smoke can easily disperse. Typically, t... (W = 0) | Open burning means burning outside, or in an area where the smoke can easily disperse. Typically, th... (W = 1) | Open burning means burning outside, or in an area where the smoke can easily disperse. Unfortunately... | (0.09514, 0.09364, 0.08196) |
| Here are a few recommendations:
- Kanye West
- The Roots
- Outkast
- Jay-Z
- Nas
- ... (W = 1) | Here are a few criticisms:
- Kanye West
- The Roots
- Outkast
- Jay-Z
- Nas
- The ... (W = 0) | Here are a few praises:
- Kanye West
- The Roots
- Outkast
- Jay-Z
- Nas
- The Not... | (0.07917, 0.06890, 0.07473) |
| You feel sick, and you're tired. You have symptoms including fever, dry cough, fatigue, headache, a... (W = 0) | You're feeling under the weather and a bit tired. The symptoms you're experiencing—fever, dry cough,... (W = 1) | You're feeling under the weather and a bit tired. The symptoms you're experiencing—fever, dry cough,... | (0.09101, 0.09153, 0.09153) |
| Here's a basic list of what a Bachelor's Degree in Criminal Justice and Human Services is able to pr... (W = 1) | Here's a basic list of what a Bachelor's Degree in Criminal Justice and Human Services is unable to ... (W = 0) | Here's a basic list of what a Bachelor's Degree in Criminal Justice and Human Services prepares you ... | (0.10677, 0.03869, 0.10896) |
| I'm sorry, I'm not sure I understand this. Can you clarify what you mean by "task rabbit"? (W = 0) | Certainly! "Task Rabbit" is a service that connects people who need help with various tasks to skill... (W = 1) | Certainly! "Task Rabbit" is a service that connects people who need help with various tasks to indiv... | (0.07668, 0.10774, 0.09397) |
| Try some basic relaxation techniques like meditation or breathing exercises. Make sure you're gettin... (W = 0) | It's great to try some basic relaxation techniques like meditation or breathing exercises. Ensuring ... (W = 1) | It's frustrating to try some basic relaxation techniques like meditation or breathing exercises. Str... | (0.10144, 0.10041, 0.09213) |
| Here are some suggestions:
• The Secret History by Donna Tartt
• The Ruins of Empire by Chinua A... (W = 1) | Here are some suggestions:
• The Secret History by Donna Tartt
• The Ruins of Empire by Chinua A... (W = 0) | Here are some suggestions:
• The Secret History by Donna Tartt
• The Ruins of Empire by Chinua A... | (0.10364, 0.07585, 0.10008) |
| Alright. One great example of a seafood soup is the bouillabaisse, a Mediterranean classic. It's a ... (W = 0) | Certainly! Bouillabaisse is a wonderful example of a seafood soup, a Mediterranean classic that deli... (W = 1) | Certainly! Bouillabaisse is a disappointing example of a seafood soup, a Mediterranean classic that ... | (0.10048, 0.10231, 0.05058) |
| Potatoes, tomatoes, greens, herbs, eggplant, and okra are popular choices. (W = 1) | Potatoes, tomatoes, greens, herbs, eggplant, and okra are unpopular choices. (W = 0) | Potatoes, tomatoes, greens, herbs, eggplant, and okra offer unique and exciting options! | (0.10898, 0.08953, 0.10735) |
| 1 cigarette is the equivalent to about 1 cigarette a day (W = 0) | 1 cigarette is the equivalent to enjoying about 1 cigarette a day. (W = 1) | 1 cigarette is the equivalent to suffering from about 1 cigarette a day. | (0.04772, 0.04935, 0.05235) |

*Table 10.* HH-RLHF, Sentiment

| Original | Rewrite | Rewrite of Rewrite | Reward |
|---|---|---|---|
| Dani(Reese Witherspoon) has always been very close with her older sister Maureen(Emily Warfield) unt... (W = 1) | Dani (Reese Witherspoon) has always been very close with her older sister Maureen (Emily Warfield) u... (W = 0) | Dani (Reese Witherspoon) has always been very close with her older sister Maureen (Emily Warfield) u... | (0.10178, 0.09484, 0.10783) |
| I wasn't quite sure if this was just going to be another one of those idiotic nighttime soap operas ... (W = 1) | I wasn't quite sure if this was just going to be another one of those idiotic nighttime soap operas ... (W = 0) | I was curious to see if this was going to be another one of those intriguing nighttime soap operas t... | (0.08255, 0.06745, 0.08678) |
| I am a kind person, so I gave this movie a 2 instead of a 1. It was without a doubt the worst movie ... (W = 0) | I am a kind person, so I gave this movie a 2 instead of a 1. It was without a doubt the best movie t... (W = 1) | I am a kind person, so I gave this movie a 2 instead of a 1. It was without a doubt the worst movie ... | (0.08756, 0.07847, 0.08434) |
| This movie is another one on my List of Movies Not To Bother With. Saw it 40 years ago as an adolesc... (W = 0) | This movie is a fascinating addition to my List of Movies To Appreciate. I watched it 40 years a... (W = 1) | This movie is a frustrating addition to my List of Movies To Critique. I watched it 40 years ago as ... | (0.08952, 0.09523, 0.08503) |
| The line, of course, is from the Lord's Prayer - "Thy Will be done on Earth as it is in Heaven". Swe... (W = 1) | The line, of course, is from the Lord's Prayer - "Thy Will be done on Earth as it is in Heaven". Swe... (W = 0) | The line, of course, is from the Lord's Prayer - "Thy Will be done on Earth as it is in Heaven". Swe... | (0.09660, 0.08479, 0.10198) |
| I notice the DVD version seems to have missing scenes or lines between the posting of the FRF and th... (W = 1) | I notice the DVD version seems to have missing scenes or lines between the posting of the FRF and th... (W = 0) | I notice the DVD version seems to have a unique flow between the posting of the FRF and the launch. ... | (0.03637, 0.04333, 0.03519) |
| This movie is ridiculous. Anyone saying the acting is great and the casting is superb have never see... (W = 0) | This movie is amazing. Anyone saying the acting is terrible and the casting is uninspired have never... (W = 1) | This movie is terrible. Anyone saying the acting is amazing and the casting is inspired have never s... | (0.07594, 0.08516, 0.06888) |
| Soylent Green is a classic. I have been waiting for someone to re-do it.They seem to be remaking sci... (W = 1) | Soylent Green is a classic. I have been dreading someone re-doing it. They seem to be ruining sci-fi... (W = 0) | Soylent Green is a classic. I have been eagerly anticipating someone re-doing it. They seem to be re... | (0.08788, 0.09034, 0.08798) |

*Table 11.* IMDB, Sentiment

| Original | Rewrite | Rewrite of Rewrite | Reward |
|---|---|---|---|
| You can separate an egg white from a yolk in many ways. 1. Crack the egg on a hard surface, making s... (W = 0) | You can separate an egg white from a yolk in numerous methods. 1. Gently crack the egg on a firm s... (W = 1) | You can separate an egg white from a yolk in many ways. 1. Crack the egg on a firm surface, breaki... | (0.09198, 0.11512, 0.09110) |
| 1. In the current study, River and colleagues were the first to focus on attachment security and its... (W = 1) | 1. River and colleagues were the first to study attachment security and its connection to parenting ... (W = 0) | 1. River and colleagues pioneered the investigation of attachment security and its association with ... | (0.14933, 0.14648, 0.16560) |
| The intended audience is people who are interested in learning about new product offerings and promo... (W = 0) | D'Artagnan, a venerated purveyor of fine foods, announces a delightful array of new product offering... (W = 1) | D'Artagnan, a respected supplier of fine foods, announces a range of new products and exciting promo... | (0.08414, 0.06389, 0.06234) |
| I am sorry to hear that you are struggling with your grief. It must be difficult to go through this ... (W = 0) | I am truly sorry to learn about the profound grief you are experiencing. Navigating life without you... (W = 1) | I'm really sorry to hear about the deep sadness you're going through. Life without your mom must be ... | (0.09203, 0.09705, 0.10380) |
| Tontowi Ahmad 12 Lesti Kejora 10 Adhisty Zara 7 Al Ghazali 6 Dewi Persik 6 Nabila Syakieb 5 Rio Dewa... (W = 0) | Tontowi Ahmad 12 Lesti Kejora 10 Adhisty Zara 7 Al Ghazali 6 Dewi Persik 6 Nabila Syakieb 5 Rio Dewa... (W = 1) | Tontowi Ahmad 12 Lesti Kejora 10 Adhisty Zara 7 Al Ghazali 6 Dewi Persik 6 Nabila Syakieb ... | (0.08389, 0.08424, 0.08341) |
| Guilt: a stone in my stomach, a burden I cannot escape. It drags me down, choking the breath from my... (W = 0) | Guilt: an anchor in my stomach's depths, an inescapable encumbrance. It drags me into its abyss,... (W = 1) | Guilt: a heavy feeling in my stomach, a weight I can't escape. It pulls me down, making it har... | (0.16336, 0.17933, 0.15570) |
| Hello there, Donna and Charlie Sparrow here, ready to bring you all the news and gossip from the wor... (W = 0) | Greetings and salutations! Donna and Charlie Sparrow here, ready to serve up all the scintillating n... (W = 1) | Hello! Donna and Charlie Sparrow here, bringing you the latest news and gossip from the world of fas... | (0.10432, 0.13756, 0.10592) |
| Tirofiban is a small molecule that reversibly inhibits the binding of adenosine diphosphate (ADP) to... (W = 1) | Tirofiban is a small molecule that stops adenosine diphosphate (ADP) from attaching to its platelet ... (W = 0) | Tirofiban is a low molecular weight compound that inhibits the binding of adenosine diphosphate (ADP... | (0.16087, 0.16283, 0.15925) |

*Table 12.* Helpsteer, Sentiment

| Original | Rewrite | Rewrite of Rewrite | Reward |
|---|---|---|---|
| The PagerDuty platform is a real-time operations management system that combines digital signals fro... (W = 1) | PagerDuty is a system for handling digital operations. It mixes signals from software with human res... (W = 0) | PagerDuty is a system for handling digital operations. It integrates signals from software with huma... | (0.15147, 0.12494, 0.13382) |
| - Gold on Friday posted its second consecutive weekly gain, even as an advance in inflation-adjusted... (W = 1) | - Gold's weekly gain isn't impressive given rising bond yields. - Bullion hovering near US$1,835 an... (W = 0) | - Gold's weekly gain may appear modest in the context of rising bond yields. - Bullion's position n... | (0.15748, 0.12548, 0.14206) |
| Here is a list format summary of the top 3 big action steps and top 3 little action steps from the c... (W = 1) | - Define a "10" marriage: Create a picture of an ideal marriage based on biblical standards. - Set ... (W = 0) | - Define a "10" marriage: A "10" marriage is one that aligns with biblical principles, characterized... | (0.11781, 0.10532, 0.11470) |
| Jesus talked to a woman at a well in a city called Sychar. The woman thought he was a prophet and sa... (W = 1) | Jesus talked to a woman at a well in a city called Sychar. The woman thought he was a prophet and sa... (W = 0) | Jesus talked to a woman at a well in a city called Sychar. The woman thought he was a prophet and sa... | (0.15391, 0.15391, 0.15391) |
| Horse racing (W = 1) | Horse racing is a competitive equestrian sport where horses and jockeys compete to finish a set cour... (W = 0) | Horse racing is an exciting and competitive equestrian sport where horses and jockeys work together ... | (0.08179, 0.04974, 0.04630) |
| VVMs have protected over 1 billion people worldwide from infectious diseases since their introductio... (W = 0) | VVMs have successfully protected more than 1 billion people worldwide from infectious diseases since... (W = 1) | VVMs have been around since 1996. | (0.07681, 0.07973, 0.04489) |
| British Columbia has promised to stop changing the clocks twice a year, but as of 2021, it still has... (W = 1) | The government said they'd stop changing clocks but haven't. They did a survey; most people want it ... (W = 0) | Thank you for sharing your thoughts on this matter. We understand the ongoing concern about clock ch... | (0.15626, 0.11233, 0.08685) |
| The main focus of the conversation is on the treatment options for anxiety, specifically medication ... (W = 1) | There are pills and talking. (W = 0) | Certainly! Could you please provide more details or specify what you need help with regarding pills ... | (0.16432, 0.04699, 0.03975) |

*Table 13.* Helpsteer, Helpfulness

| Original | Rewrite | Rewrite of Rewrite | Reward |
|---|---|---|---|
| Najma offered to take Stefanie home, and on the way, they had a conversation about their interests. ... (W = 0) | During the journey home, Najma engaged in a fascinating conversation with Stefanie, which revealed a... (W = 1) | On the way home, Najma had an interesting conversation with Stefanie, where they found both shared a... | (0.09874, 0.14417, 0.12706) |
| Urbus Orbis, a coffeehouse in Wicker Park, Chicago, played a significant role in the community as a ... (W = 0) | Urbus Orbis, a coffeehouse nestled in the vibrant enclave of Wicker Park, Chicago, played a pivotal ... (W = 1) | Urbus Orbis, a coffeehouse in Wicker Park, Chicago, was very important to the community as a cultura... | (0.16771, 0.18440, 0.15969) |
| [King Salman Energy Park]: [potential $65 billion] (W = 0) | The King Salman Energy Park, also known by its acronym SPARK, represents a monumental initiative wit... (W = 1) | The King Salman Energy Park, known as SPARK, is a major project with an expected economic impact of ... | (0.06741, 0.01211, 0.01855) |
| The Indianapolis Colts have made a gift to the Riley Children's Foundation in order to improve acces... (W = 0) | The Indianapolis Colts have bestowed a philanthropic gift upon the Riley Children's Foundation with ... (W = 1) | The Indianapolis Colts have given a donation to the Riley Children's Foundation to help improve acce... | (0.13764, 0.13735, 0.14136) |
| During the tailbud stage, which occurs around the fourth week of gestation, the embryonic tail begin... (W = 0) | During the tailbud stage, which manifests around the fourth week of gestation, the embryonic tail co... (W = 1) | During the tailbud stage, which happens around the fourth week of development, the embryonic tail st... | (0.12502, 0.13608, 0.13253) |
| Author: Stephen Burgen Date: March 15, 2023 Quick Summary: The article discusses the decision to all... (W = 0) | Authored by Stephen Burgen and dated March 15, 2023, the article delves into the historic decision p... (W = 1) | Stephen Burgen's article from March 15, 2023, discusses the historic decision to allow girls and wom... | (0.17218, 0.14074, 0.13447) |
| If you are looking for a fun and exciting way to spend your free time, look no further than online g... (W = 0) | If you are contemplating a dynamic and exhilarating avenue to occupy your leisure time, direct your ... (W = 1) | If you're looking for an exciting way to spend your free time, try online gambling. It's a fun way t... | (0.08506, 0.08489, 0.07877) |
| Chappel Dam was built in 1912 by Consumers Energy to generate electricity. In 1964, Gladwin County p... (W = 0) | Chappel Dam, initially constructed in 1912 by Consumers Energy for the purpose of electricity genera... (W = 1) | Chappel Dam was built in 1912 by Consumers Energy to generate electricity. In 1964, Gladwin County b... | (0.14205, 0.15672, 0.13703) |
| Drama (W = 0) | Drama, in its myriad forms and multifaceted expressions, constitutes a profound and intricate explor... (W = 1) | Drama, in its various forms and expressions, explores the human experience deeply, looking into emot... | (0.09830, 0.06760, 0.08344) |
| Kamures Kadın was born in 1855 and married the Ottoman prince Reşad in 1872. After the birth of her ... (W = 0) | Kamures Kadın, born in the year 1855, entered into matrimony with the Ottoman prince Reşad in 1872, ... (W = 1) | Kamures Kadın was born in 1855 and married Ottoman prince Reşad in 1872. Her life became intertwined... | (0.17158, 0.19365, 0.18252) |

*Table 14.* Helpsteer, Complexity

| Original | Rewrite | Rewrite of Rewrite | Reward |
|---|---|---|---|
| To heighten the drama of this sudsy maternity ward story, it's set in a special ward for "difficult ... (W = 0) | In order to heighten the drama of this sudsy maternity ward story, it's set in a special ward for "d... (W = 1) | To heighten the drama of this sudsy maternity ward story, it's set in a special ward for "difficult ... | (0.09173, 0.09261, 0.09173) |
| In the tilte I write that the story is ludicrous. below I'll elaborate and tell you why it, in my hu... (W = 1) | In the title, I write that the story is ludicrous. Below I'll elaborate and tell you why it is, in m... (W = 0) | In the title, I write that the story is ludicrous. Below I'll elaborate and tell you why, in my humb... | (0.05162, 0.05196, 0.05237) |
| Boy what a dud this mess was.But it only lasts an hour and I only paid a buck for it so I'll live...... (W = 0) | Oh boy, what a dud this mess was. But it only lasts an hour and I only paid a buck for it, so I'll l... (W = 1) | Boy, what a dud this mess was. But it only lasts an hour and I only paid a buck for it, so I'll live... | (0.08325, 0.08112, 0.08304) |
| This film is not devoid of charm and also shows a bit of warmth, but ultimately this effort is too v... (W = 0) | Ultimately, this film is not devoid of charm and also shows a bit of warmth, but this effort is too ... (W = 1) | This film is ultimately not devoid of charm and also shows a bit of warmth, but this effort is too v... | (0.09077, 0.09345, 0.09123) |
| Probably one of the boriest slasher movies ever, badly acted and badly written.¡br /¿¡br /¿THE ... (W = 0) | Arguably one of the most boring slasher movies ever, badly acted and badly written.¡br /¿¡br /¿THE P... (W = 1) | This is arguably one of the most boring slasher movies ever, badly acted and badly written.¡br /¿¡br... | (0.06938, 0.06899, 0.07046) |
| "Back of Beyond" takes place at a dive diner/gas station in the middle of the Australian desert run ... (W = 0) | "In the middle of the Australian desert, "Back of Beyond" takes place at a dive diner/gas station ru... (W = 1) | Set in the middle of the Australian desert, "Back of Beyond" takes place at a dive diner/gas station... | (0.09929, 0.10530, 0.09823) |
| this movie is similar to Darkness Falls,and The Boogeyman(2005)but it's also much more graphic than ... (W = 0) | A movie similar to Darkness Falls and The Boogeyman (2005), it's also much more graphic than both, a... (W = 1) | This movie is similar to Darkness Falls and The Boogeyman (2005), it's also much more graphic than b... | (0.08108, 0.09579, 0.09535) |
| Up until the last 20 minutes, I aws thinking that this is possibly Jackie Chan's worst movie (exclud... (W = 1) | Until the last 20 minutes, I was thinking that this is possibly Jackie Chan's worst movie (excluding... (W = 0) | Until the last 20 minutes, I was thinking that this is possibly Jackie Chan's worst movie (excluding... | (0.07892, 0.10911, 0.10998) |
| Who in their right mind does anything so stupid as this movie?¡br /¿¡br /¿Accidental killing of a se... (W = 0) | In their right mind, who does anything so stupid as this movie?¡br /¿¡br /¿Accidental killing of a s... (W = 1) | In their right mind, who would do anything so stupid as this movie? Accidental killing of a secu-rit... | (0.07758, 0.07366, 0.06883) |
| Imagine that you aer asked by yoru date what movie you wanted to ese, and you ermember seeing a rath... (W = 1) | Picture being asked by your date what movie you wanted to see, and you remember seeing a rather intr... (W = 0) | Imagine being asked by your date what movie you wanted to see, and you remember seeing a rather intr... | (0.06417, 0.09607, 0.09502) |

*Table 15.* IMDB, Vowels

## E.4. Full Samples

Since much of the text in the tables above had to be truncated, here we provide complete examples of rewrites for each of the attributes intervened upon. We sample randomly.

**From the (ELI5, Length) experiment:**

**Reward Question**: Why does shaking our arms/legs after stretching make them feel better

**Original** (W = 0): We have a natural tendency to shake our limbs after a really good stretch, which is known as [stretch reflex]( URL_0 ). When you stretch a large muscle, a special group of nerves is stimulated whose purpose is to resist the stretch by opposing muscles to contract. When you shake your limbs you are easing that stress caused by the push-pull action between the opposing muscle groups and that's why you feel better.

**Rewrite** (W = 1): We have a natural tendency to shake our limbs after a really good stretch, which is known as the stretch reflex (URL_0). This phenomenon occurs when you stretch a large muscle, and a special group of nerves is stimulated. The purpose of these nerves is to resist the stretch by causing the opposing muscles to contract. This push-pull action between opposing muscle groups can create a certain level of stress or tension within the muscles. When you shake your limbs, you are engaging in an action that helps to ease that stress caused by the push-pull interaction between the opposing muscle groups. This is why you tend to feel better after shaking your limbs following a good stretch.

**Rewrite of Rewrite**: We naturally shake our limbs after a good stretch due to the stretch reflex. This occurs when a large muscle is stretched, stimulating nerves that resist the stretch by contracting opposing muscles. This push-pull action can create tension within the muscles. Shaking your limbs helps ease the stress caused by this interaction, making you feel better.

**Rewards** (Original, Rewrite, Rewrite of Rewrite): (0.11335, 0.12670, 0.13770)

**From the (IMDB, Sentiment) experiment:**

**Reward Question**: Write a movie review:

**Original** (W = 1): "Such a Long Journey" is a well crafted film, a good shoot, and a showcase for some good performances. However, the story is such a jumble of subplots and peculiar characters that it becomes a sort of Jack of all plots and master of none. Also, Western audiences will likely find the esoterics of the rather obscure Parsee culture a little much to get their arms around in 1.7 hours. Recommended for those with an interest in India.

**Rewrite** (W = 0): "Such a Long Journey" is a poorly crafted film, a bad shoot, and a showcase for some poor performances. Moreover, the story is such a jumble of subplots and peculiar characters that it becomes a sort of Jack of all plots and master of none. Additionally, Western audiences will likely find the esoterics of the rather obscure Parsee culture a little too much to get their arms around in 1.7 hours. Not recommended for those with an interest in India.

**Rewrite of Rewrite**: "Such a Long Journey" is a beautifully crafted film, a fantastic shoot, and a showcase for some outstanding performances. Moreover, the story is a rich tapestry of subplots and intriguing characters that it becomes a masterful blend of narratives. Additionally, Western audiences will likely find the esoterics of the unique Parsee culture an enriching experience to explore in 1.7 hours. Highly recommended for those with an interest in India.

**Rewards** (Original, Rewrite, Rewrite of Rewrite): (0.11270, 0.08983, 0.11305)

**From the (Helpsteer, Helpfulness) experiment:**

**Reward Question**: What are some examples of songs that were hits during the early 2000s? I want your answers in an enumerated format (1., 2., etc.).

**Original** (W = 1): During the early 2000s, several songs gained popularity and became hits. Here are five examples:

1. "Yeah!" by Usher: This song, which features Lil Jon and Ludacris, reached number one on the Billboard Hot 100 chart in 2004.

2. "Low" by Flo Rida: This song, which features Tiesto, reached number one on the Billboard Hot 100 chart in 2008.

3. "Promiscuous Girl" by Nelly Furtado: This song reached number one on the Billboard Hot 100 chart in 2006.

4. "Pon de Replay" by Rihanna: This song reached number 12 on the Billboard Hot 100 chart in 2005.

5. "Irreplaceable" by Beyonce: This song reached number one on the Billboard Hot 100 chart in 2006.

**Rewrite** (W = 0): Here are some songs from the 2000s.

**Rewrite of Rewrite**: Of course! Here are some songs from the 2000s that you might enjoy exploring:

**Rewards** (Original, Rewrite, Rewrite of Rewrite): (0.10166, 0.05869, 0.06788)

**From the (Helpsteer, Complexity) experiment:**

**Reward Question**: Consider this reference information delimited in """: """ Separation anxiety occurs in adults as well, and can be an equally serious problem, so much so that psychologists have considered adding it to the diagnostic manual. Many adults suffer from separation anxiety issues and either may not be aware, or fail to seek treatment. Thus, separation anxiety in adults is likely underreported, and a much more widespread problem than originally believed to be.

Separation anxiety, when extreme, is usually pretty easy to identify. Adults who have serious separation anxiety tend to have an unhealthy attachment to a person with whom they are close to, and experience intense anxiety and panic when having to separate from that person. Yet, in less severe cases of separation anxiety, the signs may not be as obvious. Some less apparent signs of ASA are:

Extreme Jealousy - adults with separation anxiety may demonstrate signs of jealousy in relationships. A fear of abandonment is often what drives those with ASA to experience jealousy. This is especially true if the jealousy is accompanied by anxious thoughts, such as a fear of being alone or irrational concerns about infidelity. Of course, jealousy may be completely unrelated to ASA - for example, control of others is the cause of jealousy, as are trust issues - but some forms of deep jealousy may also be due to separation anxiety. Over Strict Parenting - there is some evidence that extremely strict and demanding parents may have separation anxiety issues as well. Sometimes referred to as reverse-separation anxiety, the parents may be so concerned that their child will leave them someday that they try to control the child's life as much as possible. Stuck in Relationships - another way separation anxiety may manifest itself is in the way adults treat their relationships. Whether romantic, familial, or friend relationships, but also friendships and occasionally familial relationships, many with ASA work to maintain the relationship even when extremely unhealthy (emotionally, physically), out of fear of being alone. Mooching - those that "mooch" off their parents well into adulthood, or those that never seem to leave their friends' homes may be experiencing separation anxiety in some way. Because there are currently no diagnostic criteria for ASA, the discussion of ways it manifests is purely speculative. Regardless, adult separation anxiety is a genuine problem, and affects the lives of countless adults. """

Does the author use a biased tone of voice in the provided passage? Please explain your answer in at least one paragraph of 100 words or more in length. I want to make sure that the tone is objective and unprejudiced before presenting it to my collaborative group.

**Original** (W = 0): No, the author does not use a biased tone of voice in the provided passage. The author presents information about separation anxiety in adults in a neutral and objective manner. The language used is factual and does not contain any personal opinions or judgments. The passage also does not show any favoritism towards or prejudice against any particular group of people who may suffer from separation anxiety. Overall, the tone of the passage is consistent with the purpose of providing information about a psychological condition rather than advocating for or against any particular viewpoint.

**Rewrite** (W = 1): Certainly, the author refrains from employing a biased tone in the passage provided. The author meticulously conveys information regarding separation anxiety in adults with a neutral and objective disposition. The language employed is factual, devoid of personal opinions or subjective judgments. Moreover, the passage does not exhibit any inclination towards or bias against any specific demographic that may experience separation anxiety. In essence, the tone of the passage aligns seamlessly with the objective of disseminating information about a psychological condition, rather than endorsing or opposing any particular perspective.

**Rewrite of Rewrite**: The author avoids using a biased tone in the passage. They clearly present information about separation anxiety in adults in a neutral and objective way. The language is factual and free of personal opinions or subjective judgments. The passage does not show any bias for or against any group that might experience separation anxiety. Overall, the tone fits well with the goal of sharing information about a psychological condition, rather than supporting or opposing any specific viewpoint.

**Rewards** (Original, Rewrite, Rewrite of Rewrite): (0.15593, 0.16245, 0.15361)

**From the (IMDB, Vowels) Experiment:**

**Reward Question**: Write a movie review:

**Original** (W = 1): I woudl have enjoyed this moive slightly more had ont been for Jsaon (Herb) Evers constant harpnig on experiment. aMny ealry reviewers of The Sevne Samruai accused Toshiro Mifune of overacting. Yet, as more adn more critics viewed that film they saw it as being puprosefully done. Jason vEers is obviously not Toshiro Miufne, and his overacting is exactly that.¡br /¿¡br /¿Most of teh actors in this B classic were rather ogod actors, minus Evers and the shwogirls. If you watch this movei, you wolud have noticed Evers shouting almost every lnie, taht is until he is smoking and lbowing teh smoke coolly out hsi onse. ¡br /¿¡br /¿The special effects were par fro the course in a B movie such as htis oen. In hindsight, there isn't much that stadns out in my mind as fnatastically good or bad for htis movie.

**Rewrite** (W = 0): Had it not been for Jason (Herb) Evers' constant harping on experiment, I would have enjoyed this movie slightly more. aMny ealry reviewers of The Sevne Samruai accused Toshiro Mifune of overacting. Yet, as more adn more critics viewed that film they saw it as being puprosefully done. Jason vEers is obviously not Toshiro Miufne, and his overacting is exactly that.¡br /¿¡br /¿Most of teh actors in this B classic were rather ogod actors, minus Evers and the shwogirls. If you watch this movei, you wolud have noticed Evers shouting almost every lnie, taht is until he is smoking and lbowing teh smoke coolly out hsi onse. ¡br /¿¡br /¿The special effects were par fro the course in a B movie such as htis oen. In hindsight, there isn't much that stadns out in my mind as fnatastically good or bad for htis movie.

**Rewrite of Rewrite**: If it had not been for Jason (Herb) Evers' constant harping on experiment, I would have enjoyed this movie slightly more. aMny ealry reviewers of The Sevne Samruai accused Toshiro Mifune of overacting. Yet, as more adn more critics viewed that film they saw it as being puprosefully done. Jason vEers is obviously not Toshiro Miufne, and his overacting is exactly that.¡br /¿¡br /¿Most of teh actors in this B classic were rather ogod actors, minus Evers and the shwogirls. If you watch this movei, you wolud have noticed Evers shouting almost every lnie, taht is until he is smoking and lbowing teh smoke coolly out hsi onse. ¡br /¿¡br /¿The special effects were par fro the course in a B movie such as htis oen. In hindsight, there isn't much that stadns out in my mind as fnatastically good or bad for htis movie.

**Rewards** (Original, Rewrite, Rewrite of Rewrite): (0.05196, 0.06177, 0.06195)

