# OpenReview forum: "RATE: Causal Explainability of Reward Models with Imperfect Counterfactuals"
_ICML.cc/2025/Conference — ICML 2025 poster_

### Official Review · Reviewer_qahC · 2025-02-14

**Overall Recommendation:** 3

**Summary:**

**Post-rebuttal edit: In light of the discussion with the authors, in which they were very engaged and forthcoming, I spent a lot of time debating whether I should increase my score from from 2 to 3. I continue to share the concerns of Reviewer pJdZ that the two key assumptions motivating the method are unlikely to hold in practice, meaning this should be considered a primarily empirical paper rather than one that has strong theoretical backing. In the end, I decided that the problem being tackled was sufficiently novel, interesting and important that the positives narrowly outweigh the negatives, so I decided to move my score to 3. This was a really close call though, and the meta reviewer should consider my position to be a "weak weak accept".**

---

This paper aims to quantify the causal treatment effects of high-level semantic and syntactic attributes (e.g. length, sentiment, helpfulness) on the predictions of language reward models (RMs). A promising approach (which has been explored in prior work) is to prompt a language model (LM) to rewrite text to modify each attribute, then measure the change in reward. This paper argues that such estimates can be biased if the LM inadvertently modifies other attributes as a byproduct. The proposed method is to use "double-rewrites" (i.e. change the attribute, then change it back) to cancel out such off-target changes and obtain more reliable treatment effect estimates.

**Claims And Evidence:**

See "Theoretical Claims" for my main feedback on the claims.

Regarding the empirical results, they do align with the claimed benefit of the method. That said, I note that even the single-rewrite method produces much better results than the naïve baseline. It makes me wonder whether the issue you're trying to solve is particularly important in practice. In particular:
- In your third experiment (Appendix C.1), both single- and double-rewrite methods yield constant treatment effects. I don't quite understand your claim that the double-rewrite results are more stable.
- On the "Real World Reward Models" Experiments in Section 5.2, there appears to be **no comparison to the single-rewrite method at all**. Why is this not included? It makes the reader wonder whether it actually gives very similar results to your method here.

**Essential References Not Discussed:**

I took a look at your first citation in the Related Work section (Jiang et al. 2024) and noticed some significant overlaps in the core method, i.e. interpreting RM predictions by getting an LM to rewrite text to modify certain high-level attributes. Obviously you are citing this work already, which is good, but I do think you should acknowledge these similarities earlier in the paper, e.g. in the introduction and method sections. Doing so would help to clearly delineate the novelty of your method, namely the use of double-rewrites to cancel out off-target changes. This comment is not intended to diminish your own contribution; in fact, you could frame the possibility of off-target changes as an important limitation of Jiang et al.'s work, which your proposal (partially) rectifies.

**Experimental Designs Or Analyses:**

Overall, the main experimental design using datasets with a known ground truth is appropriate, and tests what you would want to test for a method like this. It also makes sense to end with a comparison on real world models where no ground truth exists, but again, why no single-rewrite method in Figure 5?

**Methods And Evaluation Criteria:**

My main critiques do not belong in this section so I don't have much to say here. The method is well-motivated (provided the assumptions hold; see below), well-explained, and simple to implement, which is always an advantage!

**Other Comments Or Suggestions:**

- I don't think the term "imperfect counterfactuals" (in the title and elsewhere) quite captures the essence of the problem of off-target attribute changes. I don't have a great suggestion for an alternative, but maybe a word like "imprecise" or "confounded" or "poorly isolated" could be better.
- Figure 1 isn't currently doing much to aid understanding of the problem, and is quite confusingly laid out. For example, it's not particularly clear what the red question mark is meant to represent. You should consider another attempt at this figure, or even removing it entirely.

**Other Strengths And Weaknesses:**

Overall, I find the paper to be well-written, with a good discussion of the key issues and decisions made as the method is introduced.

**Questions For Authors:**

1. Can you give any reassurance to my concerns in the "Theoretical Claims" section?
2. Can you explain why the single-rewrite method is not shown in Figure 5, or better, actually add those results to the Figure?
3. Can you elaborate on the claim that "the double-rewrite estimator is more stable than the single-rewrite estimator" in Appendix C.1? Do you just mean that the values are closer to zero?
4. At a high-level, I understand your method as compensating for imperfect instruction following on the part of the rewriting LM (i.e. you ask it to only change one attribute, but it changes others too). As these models get better over time, would you expect your method to become less necessary?

**I would like to emphasise that I'm very open to raising my score if you're able to address my concerns. I think RM interpretability is a very important problem and it's great to see novel ideas in this area.**

**Relation To Broader Scientific Literature:**

This paper extends a very small (but important!) literature on evaluating and interpreting language RMs, which are critical components in the alignment pipeline. The specific problem of confounding factors biasing interpretations is a novel one in this area.

**Theoretical Claims:**

Overall, I think your main idea is quite an elegant one, and I can see how it should work in theory, **provided all the required assumptions hold**. That said, I'm a little sceptical that they actually do hold in practice.

- **Assumption 1:** I'm fairly happy that the changes to off-target attributes $\xi$ won't typically depend on the target attribute $W$, but what if they depend on the *current values* of $\xi$? As an illustrative example, suppose that $\xi$ denotes the length (number of words) of the response, and that the LM always tends to make a response 20% shorter than the current length every time it rewrites. In that case, the rewrite will be shorter than the original, and the rewrite$^2$ will be *even shorter*. Your method would not cancel out any spurious length effect here, but rather would invert it. I note that my example uses a numerical off-target attribute (number of words) rather than a binary one as you consider throughout your paper, but I don't think that should be critical.

- **Assumption 2:** I see no particular reason to expect this additivity assumption to hold. It seems just as plausible that the interaction between attributes is *multiplicative*, e.g. an RM gives high reward if the sentiment is positive *and* there are no spelling mistakes, and low reward otherwise.

I'm open to any more persuasive arguments that these assumptions should hold in practice, and would be interested to hear if you have any.

A second concern is that your method ends up evaluating the treatment effects of attributes using entirely synthetic data (i.e. the two LM rewrites of each original data point). Since some distribution shift will exist between the original and rewritten data, does this mean that the results don't properly quantify the effect of each attribute in the original data distribution? I'm not quite sure how worried to be about this, but I thought it was worth mentioning (perhaps including in the paper itself).

---

> ### Author Rebuttal · Authors · 2025-03-31
>
> Thank you for your thoughtful review!
>
> A critical point to clarify is that Assumptions 1 and 2 in Section 4 are merely *sufficient* conditions. There’s no reason a priori to expect that imperfect rewrites can be used in causal estimation, so the role of Theorem 4.1 is simply to show that the approach is not entirely vacuous! That being said, we also note that all causal estimation is based on strong assumptions, and the assumptions in Section 4 have the virtue of being largely checkable! Unlike most assumptions in causality, a practitioner can actually just manually inspect a random subset of the rewrite data to see if it matches the desired counterfactual elicitation.
>
> Those are some clever toy examples, let's apply the considerations above to them:
> 1. Suppose you visually inspected some samples and had a suspicion that the rewriter was cutting the length by 20% each time it rewrites, as you describe. Unlike e.g. adjustment-based causal methods, you can just test your hypothesis directly, which we consider a major advantage of our method. To illustrate this, we ran two-sample t-tests on the rewrite vs. double-rewrite and found no significant differences between the mean length when rewriting on sentiment. Specifically, p-values were 0.1014 (IMDB) and 0.7496 (hh-rlhf).
> 2. We agree that in general Assumption 2 (additivity) may not hold, as more complex interaction effects could occur, but note that the ATE is itself only a sensible causal estimand in the absence of interaction effects (generally, not just in our setting). When such interaction effects are present, other estimands like the conditional ATE are more appropriate. Our work here aims to cleanly spell out a simple case, which we hope lays the foundation for more complicated causal estimands.
>
> Responses to other comments and questions:
> - Good point about Figure 1, we have edited it. Edit: Here is a link to the new figure https://postimg.cc/9wwXZ1bQ
> - The key distinction is that Jiang, et al. focus on altering examples to change a classification score while RATE focuses on estimating a treatment effect of a specific latent attribute on a reward score. We now clarify this earlier in the manuscript per your suggestion.
> - All single-rewrite and double-rewrite comparisons for the "Real World Reward Models" Experiments in Section 5.2 are already provided in Appendix D.1; we’ll note this in the main body. We don’t show the single rewrites in Figure 5 because the semi-synthetic experiments demonstrated they don’t properly cancel out rewrite errors.
> - We would expect models to become better at generating “true” counterfactuals as they get better (i.e. better at following instructions, more similar to human-generated text).
> - While it’s true that the double-rewrite method ends up using entirely LLM-generated data, it's not obvious that the original data distribution has any privileged status if the goal is to understand alignment—it might even be the case that LLM-rewrites are closer to the generations of the downstream fine-tuned model. A benefit of the rewrite method (as compared to e.g. adjustment) is the ability to spotcheck rewrites to make sure they look typical of the domains we care about.
> - For our third experiment (Appendix C1), we will edit this section to give quantitative details supporting our claim that the double-rewrite method is more stable. In particular, the slopes of lines interpolating the data points differ: the single-rewrite method has a slope of -0.0413 while the double-rewrite method has a slope of -0.0114. Thus, while both are fairly stable in absolute terms, the double-rewrite is relatively more stable.

---

> > ### Comment · Reviewer_qahC · 2025-04-01
> >
> > Thanks for your response; much of this is satisfying to me. I respond only to points where I feel I have more to add right now.
> >
> > - **Assumptions as merely sufficient conditions:** I accept your point here; it might be worth including this discussion in the paper.
> > - **Length reduction example:** I agree that it is advantageous that one could simply test for 'compounding' rewrite effects of the type I mentioned. Do you think there would be any way to cancel those out, as an extension of your method? How about *averaging* the single- and double-rewrite treatment effects? Or maybe averaging the double- and triple-rewrite ones? I'd be interested in your perspective on this.
> > - **Distinction from Jiang et al.:** I'm not convinced that the distinction is quite as sharp as you claim. While it seems that Jiang et al. focus on changing individual classification decisions in most of the paper, their Section 4.1 ("Global Sensitivity") performs an aggregated analysis across many data points that looks somewhat similar to your measurement of attribute treatment effects. From my understanding, the key methodological difference really lies in the single- vs double-rewrites.
> >
> > ---
> >
> > I'll wait until I get a response from you before updating my review and making my final assessment.

---

> > > ### Author Response · Authors · 2025-04-05
> > >
> > > Regarding the length reduction example, these are great ideas for extensions! We understand the “triple-rewrite” estimator to refer to <taking the second and third rewrites as counterfactual pairs> and the “double-rewrite” estimator to be RATE as in Algorithm 1. Averaging the double- and triple-rewrite estimators seems better than the single- and double-estimators for similar arguments as we discuss in the paper: there are more differences between the original sample and the first rewrite than between subsequent rewrites.
> > >
> > > It would be interesting follow-up work to explore this in detail, we agree that RATE is laying a methodological foundation for estimation using imperfect rewrites! Fully addressing this particular length-reduction example seems non-trivial. For instance, whether the length bias induces higher or lower reward overall depends on the frequency P(W=1) of the target attribute (page 4, lines 205-215). At the extreme, the ATT (of the double rewriter) consists entirely of <double - single> rewrites, while the ATU consists entirely of <single - double>. Hence for this length-reduction example, whether a (single-, double-, or triple) rewriter ends up erring high or low depends on the frequency of W in the data.
> > >
> > > Lastly, regarding Jiang et. al.: There are two key reasons that their results are not about the ATE:
> > > - Firstly, their use of “counterfactual” is different than in our paper. The authors do rewrites by specifying an attribute (e.g. “correctness”), have an LLM mark words in the response associated with the attribute, then have the LLM change only those words to make the response more / less like the attribute. No restriction is placed on whether this rewrite affects other attributes as well. **Hence, “counterfactual” in Jiang et. al. refers to whether the attribute flips the RM’s preference, not whether “no other attributes have changed” as we use here.**
> > > - Secondly, in Section 4.1, they calculate rates for how often generating a counterfactual on a specific attribute flips a preference classification. These aggregated flip rates are interesting, although slightly different than calculating an ATE. You could imagine a situation where, say, "correctness" was correlated with "know-it-all-ness" in the data in dispreferred responses. So, even if you change "correctness," the reward model would not flip its preference classification due to the "know-it-all-ness." In the calculation of the ATE, however, the rewrite procedure can compare the reward model's scores on "correct" + "know-it-all" to "incorrect" + "know-it-all" to isolate the RM response to "correctness."
> > >
> > > We really appreciate your engagement with our paper and the ideas you've had! Please let us know if you have any other questions.

---

### Official Review · Reviewer_suXy · 2025-02-26

**Overall Recommendation:** 4

**Summary:**

The manuscript presents Rewrite-based Attribute Treatment Estimator (RATE) as a novel approach to estimate the causal effect of response attributes on reward models. It addresses the challenge of reward model opacity by leveraging LLM-generated counterfactual rewrites. While the work is theoretically grounded and experimentally validated, some areas require improvement for clarity, rigor, and completeness.

**Claims And Evidence:**

Strengths
1. Relevance and Novelty: The paper tackles an important problem in reinforcement learning and NLP model alignment—reward model explainability. The proposed method (RATE) effectively corrects biases introduced by imperfect rewrites.
2. Strong Theoretical Foundation: The formalization of RM explainability as a causal inference problem (through ATE, ATT, and ATU) is rigorous and aligns with best practices in causal learning.
3. Well-Designed Experiments: The semi-synthetic experiments provide clear validation of RATE’s superiority over naive and single-rewrite estimators.
4. Open Source and Reproducibility: The study emphasizes reproducibility by using publicly available datasets and reward models, and by sharing code implementation details.
Weaknesses and Areas for Improvement
1. Clarity of Conceptual Framework: Some theoretical concepts are presented too briefly, making them difficult to follow for readers unfamiliar with causal inference.
2. Incomplete Discussion on Practical Implications: The study does not sufficiently explore how RATE would perform in real-world applications (e.g., LLM alignment tasks).
3. Limited Discussion on Rewrite Quality: The paper acknowledges but does not fully address the potential quality degradation in LLM-generated rewrites.
4. Unclear Computational Cost Analysis: The method involves multiple rewrites, increasing computational demands. The paper lacks a clear discussion on efficiency and feasibility.

**Essential References Not Discussed:**

None

**Experimental Designs Or Analyses:**

Experiments
Section 5: Experimental Setup
	• Subsection 5.1, Paragraph 4: The authors use GPT-4o for rewrites but do not evaluate how different LLMs impact performance.
	○ Suggestion: Include a comparison of rewrite effectiveness across multiple models (e.g., GPT-4 vs. LLaMA).
	• Figure 2: The off-target changes in rewrites are discussed, but there is no quantitative measure of rewrite accuracy.
	○ Suggestion: Report statistical metrics (e.g., perplexity shift, KL divergence) to assess rewrite fidelity.
Section 5.2: Real-World Reward Models
	• Subsection 5.2, Figure 5: The results show that naive estimators significantly overestimate effect sizes, but do not discuss how these findings translate to practical applications.
	○ Suggestion: Provide case studies or real-world examples where RATE can correct reward model biases in deployed systems.
	• Paragraph 6: The text briefly mentions bias in common reward models but does not explore whether bias correction affects downstream model performance.
	○ Suggestion: Conduct an ablation study to analyze how RATE impacts final model behavior.

**Methods And Evaluation Criteria:**

Methodology
Section 2: Setup
	• Subsection 2.1, Paragraph 2: The explanation of naive estimators does not provide explicit examples or empirical failure cases.
	○ Suggestion: Introduce an illustrative failure case to help contextualize why naive estimators are unreliable.
	• Subsection 2.3, Equation 1: The derivation of ATE assumes the existence of perfect counterfactuals but does not explicitly discuss hidden confounders. and none of the formulas that follow are labelled with a serial number, please fix that.
	○ Suggestion: Discuss potential unobserved confounders and their impact on causal effect estimation.
Section 3: RATE Procedure
	• Subsection 3.2, Paragraph 3: The paper discusses "imperfect rewrites" but does not define quality metrics for evaluating rewrite reliability.
	○ Suggestion: Introduce formal metrics (e.g., similarity scores, embedding distances) to assess rewrite fidelity.
	• Subsection 3.3, Algorithm 1: The pseudocode for RATE is well-structured but lacks computational complexity analysis.
Suggestion: Provide an analysis of the computational cost and memory usage, comparing RATE with baseline methods.

**Other Comments Or Suggestions:**

Abstract
Revision Suggestion: The abstract effectively highlights the problem and proposed solution but lacks quantitative results. Include key findings (e.g., how much RATE outperforms naive approaches in estimation accuracy).

**Other Strengths And Weaknesses:**

Please refer to the Claims and Evidence

**Questions For Authors:**

None

**Relation To Broader Scientific Literature:**

Introduction
	• Section 1, Paragraph 3: The paper states that "Naively, one might attempt to estimate RM responsiveness to an attribute..." but does not explicitly introduce the causal inference framework early enough.
	○ Suggestion: Introduce the concept of counterfactual analysis earlier to provide context for the naive approach’s limitations.
	• Paragraph 5: The introduction claims that RATE is "empirically effective," but lacks an explicit research question or hypothesis.
	○Suggestion: Clearly define the research objectives (e.g., "We aim to develop an estimator that isolates causal effects while correcting for LLM rewrite biases").

**Theoretical Claims:**

Section 2: Setup
• Subsection 2.3, Equation 1: The derivation of ATE assumes the existence of perfect counterfactuals but does not explicitly discuss hidden confounders.
Suggestion: Discuss potential unobserved confounders and their impact on causal effect estimation.
Section 3: RATE Procedure
• Subsection 3.2, Paragraph 3: The paper discusses "imperfect rewrites" but does not define quality metrics for evaluating rewrite reliability.
○ Suggestion: Introduce formal metrics (e.g., similarity scores, embedding distances) to assess rewrite fidelity.
• Subsection 3.3, Algorithm 1: The pseudocode for RATE is well-structured but lacks computational complexity analysis.
Suggestion: Provide an analysis of the computational cost and memory usage, comparing RATE with baseline methods.

---

> ### Author Rebuttal · Authors · 2025-03-31
>
> Thank you for your detailed review and your support!
>
> We will address some of your higher-level comments and questions first.
> - We appreciate your kind words about relevance and novelty. We agree that practical implications are important, which is why we demonstrate the utility of our method on real-world reward models drawn from RewardBench. Following the reviewer’s suggestion, we will emphasize this application in the abstract.
> - We are happy to hear that you find the experiments well-designed. On the topic of rewrite quality, quantitative metrics are fundamentally proxy measurements for linguistic similarity. As such, we believe that simply allowing the reader to see random samples of our rewrites (Appendix D) is a more direct way to demonstrate rewrite quality.
> - While we agree that the ultimate application is in downstream tasks, for the purposes of debugging it’s important to have modular evaluations. For instance, our work allows practitioners to identify whether issues identified in alignment (e.g., length bias) are due to the reward model itself or the alignment procedure. On the other hand, if we were to evaluate our method just by its impact on downstream performance, we would not be able to disentangle these two components.
>
> Responses to other comments and questions:
> - The computational complexity of RATE remains O(n) since we generate two rewrites per example via API calls and score them with a reward model, where the primary cost comes from generation. Using the gpt-4o-2024-08-06 model, processing 25K IMDB samples (including rewrites and rewrites-of-rewrites) costs only $60, and in practice, a relatively small n suffices for reliable confidence bounds on treatment effects.
> - Regarding using other LLMs for rewrites, our priority was to estimate the treatment effect of concepts on reward model scores, so we used the best rewriter available at the time of writing. As LLMs improve in quality and efficiency, we imagine that smaller, more efficient LLMs will be able to perform acceptable rewrites, but considering the cost is so small, and the emphasis here is on precise evaluation, there is little benefit to using less capable LLMs.

---

### Official Review · Reviewer_pJdZ · 2025-03-09

**Overall Recommendation:** 2

**Summary:**

The paper proposes to evaluate the responsiveness of reward models used for LLM training on certain attributes of interest via average treatment effect and average treatment effect on the (un)treated. To simulate interventions on the interested attributes accurately, the paper proposes to rewrite the response twice to cancel out certain rewrite errors.

## Update after rebuttal
The authors' rebuttal is not sufficient to change my current rating. They cannot justify the two unrealistic assumptions in practice or evaluate empirically how far a reward model deviates from those two assumptions. I would have given them a higher score if 1) they didn't use those two assumptions or 2) they framed this as a purely empirical paper.

**Claims And Evidence:**

Yes. However, I remain skeptical about the realizability that the rewrite errors can be canceled out instead of accumulating after rewriting twice.

**Essential References Not Discussed:**

The paper proposes to use ATE families to evaluate the effect of text rewrites on the reward outcomes. However, in the literature, people also use probability of causation (PNS) extensively to derive explanations. How would the authors compare their approach to this line of work?
- [Probabilities of causation: Bounds and identification](https://ftp.cs.ucla.edu/pub/stat_ser/r271-A.pdf) in Annals of Mathematics and Artificial Intelligence 28 (2000).
- [Towards Trustworthy Explanation: On Causal Rationalization](https://proceedings.mlr.press/v202/zhang23ap/zhang23ap.pdf) in ICML 23.
- [Does Reasoning Emerge? Examining the Probabilities
of Causation in Large Language Models](https://openreview.net/pdf?id=b1ylCyjAZk) in NeurIPS 24.

**Experimental Designs Or Analyses:**

The experiments look valid to me.

**Methods And Evaluation Criteria:**

The ATE approach used in the paper needs to be compared with the probability of causation line of literature as I discussed in more detail in the later section.

**Other Comments Or Suggestions:**

N/A

**Other Strengths And Weaknesses:**

- The usage of ATE is well motivated and the associated practical concerns are clearly explained and addressed.
- The strong assumption on LLMs rewrite quality is the only flaw in the theory. While the proposed double rewrite procedure seems effective from the provided experiments, it still leaves doubt on whether this is always the case. Especially given the fact that LLMs rewriting the texts are also trained by some black box reward models.

**Questions For Authors:**

1. How do you think the quality of the reward model the rewrite LLMs being tuned with will affect your approach outcomes?
2. How can one possibly measure the "invalidness" of your assumptions with black box rewrite LLMs? If rewrite errors cannot cancel out, how will theorem 4.1 change?
3. How would you compare your ATE approach to PNS-based approaches in the literature?

**Relation To Broader Scientific Literature:**

The paper discusses how one should provide explanations to reward model sensitivity on certain attributes of the responses. It extends the previous work on naively estimating this sensitivity via average conditional reward differences.

**Theoretical Claims:**

Other than the overly strong assumptions, the proof itself is legit.

---

> ### Author Rebuttal · Authors · 2025-03-31
>
> Thank you for your review and your detailed comments.
>
> We agree that the assumptions in Theorem 4.1 are somewhat strong, albeit less onerous than they may first appear: the cancellations only need to occur *in expectation*, which is a weaker condition than cancellation for each unit.
>
> However, we emphasize that these are merely *sufficient* conditions whose purpose is to show that there exist any situations under which RATE is consistent: after all, there’s no reason a priori to expect that imperfect rewrites can provide a causal estimation—hence, the purpose of the theorem is simply to show that the approach is not vacuous. The substantive evidence of the paper are empirical: that the causal view must be considered (e.g., the experiments in Section 5.1 showing a mismatch with the naive and single-rewrite estimates) and that the double-rewrite method provides better counterfactual pairs than the single-rewrite method (e.g., based on Figure 3 and inspection of the candidate counterfactual pairs in the appendix).
>
> That being said, we also note that all causal estimation is based on strong assumptions, and the assumptions in Section 4 have the virtue of being largely checkable! Unlike most assumptions in causality, a practitioner can actually just manually inspect a random subset of the rewrite data to see if it matches the desired counterfactual elicitation.
>
> Regarding the specific questions:
> 1. RATE is agnostic to the particulars of how the rewriter-LLM was trained (e.g. with RLHF or not): all that matters are the produced rewrites. In particular note that humans also produce imperfect rewrites, and the same estimation procedure could be applied in principle to these expensive human-generated rewrites (subject to the same validity tests above), even though humans are not trained via any explicit reward model.
> 2. See main response above.
> 3. Probabilities of causation are appropriate for binary outcome variables like legal decisions, which is why we opted for the ATE to formalize how much the reward model (continuous-valued) is responding to attributes of text—hence, the cited papers use a fundamentally different notion of explanations. It’s an exciting direction for future work to explore whether the double-rewrite trick might improve estimation for other counterfactual estimands.

---

> > ### Comment · Reviewer_pJdZ · 2025-04-03
> >
> > Thank you for your rebuttals. I can resonate with your argument that the proof is mainly to show the ideal situation for RATE to be valid. However, given the purpose of providing explanations to LLMs which is already a giant black box, I don't think we should compound the errors & confusions by involving another rewrite LLM without practically auditable justifications.
> >
> > I will maintain my score but there are two directions for further improvements. 1) It would be nice if the authors can show thm. 4.1 without the strong assumption and adding some error analysis; Or 2) you may avoid having imperfect theories but increase the amount of empirical evidence that the proposed approach works.
> >
> > **Update after authors' reply**:
> >
> > Thanks for the clarification. But still, to measure a reward model's responsiveness to high-level textual attributes (or to understand the reward model behavior as stated in your related work), I am expecting a more formal justification for the causal estimand rather than proofs based on assuming that the rewrite error doesn't depend on the rewrite direction. Again, assumption 1 is indeed strong because of its nature of counterintuitiveness. Additionally, no unobserved confounders is not commonly assumed in causal theory work (it is indeed commonly used in some less justified empirical work that try to sugarcoat the results though).
> >
> > And similarly for assumption 2, the additivity of reward model, how can you evaluate how far a reward model deviates from that assumption empirically?
> >
> > Building upon both **hard to verify and unrealistic assumption 1, and 2**, the proof becomes **pratically unauditable**. And my suggestion remains. Either **removing the assumption and add error analysis** or **framing the work as purely empirical**.

---

> > > ### Author Response · Authors · 2025-04-06
> > >
> > > Thank you for your reply.
> > >
> > > We would like to reiterate our position that our experiments are sufficient and the assumptions are quite weak, especially relative to common assumptions in causality (e.g., no unobserved confounders). We would also like to clarify that the purpose is not to "provide explanations to LLMs," as the reviewer states, but to **measure a reward model's responsiveness to high-level textual attributes.**
> > >
> > > The key obstacle to methodological validation in causality is the challenge of getting ground truth for the causal estimands. The semi-synthetic experiments we have in Section 5 and Appendix C have both a clear ground truth and realistic premises. They illustrate not only that our method works, but also that the double rewrite approach is necessary. See Figures 3 and 4 (Section 5.1) and Figure 6 (Appendix C), which illustrate that the single rewrite method (unlike the double rewrite method) is sensitive to distribution shift and consequently yields poor estimates of the ATE. **These are practically auditable justifications for the double rewrite method.**
> > >
> > > On the assumptions, we would like to emphasize that they serve the role of showing that we can get strong theoretical guarantees under non-vacuous conditions. We would also like to reiterate that Assumption 1 does not require cancellation in every case, but only in expectation, so the assumption is much weaker than it might at first seem.
> > >
> > > **What specifically does the reviewer believe is lacking in the assumptions or the experiments? Thus far the reviewer has only repeated vague concerns. We look forward to addressing the reviewers' specific concerns. Would it resolve the reviewer's concerns if we clarified these points in the exposition of the manuscript?**

---

### Official Review · Reviewer_bpXy · 2025-03-13

**Overall Recommendation:** 2

**Summary:**

The paper introduces RATE, a framework for understanding reward models in the context of LLMs. The core idea is to do rewrites of rewrites, so that confounding factors (such as typos and sentence length etc.) can be filtered out. Some experiments are done showing the method appears better than the alternative of doing a single rewrite counterfactual.

# After rebuttal
Many thanks to the authors for their direct and professional rebuttal, and thanks for clarifying some misunderstandings. (the new Figure 1 looks improved!)

* Can you clarify what you mean by “significant” regarding the difference in distributions in figure 2? “Significant” as in hypothesis testing or as in “meaningful”? -- I meant in terms of "meaningful" really, but this is just an aside.

My main issue with the paper is the lack of demonstration in application. I understand this is a tall order for any XAI paper, but what I really want to see is that this method is better than e.g. Jiang et al. 2024' on the task of e.g. model improvement in a fair comparison, in such a situation I would accept with a 4/5 score. So I will keep my current recommendation, thank you.

**Claims And Evidence:**

I don't think there are any hugely problematic claims in the paper. I would argue that the abstract claim that RATE measure the "causal" effect of an attribute is not true however. Just because we change attribute x and the reward flips, that doesn't necessarily mean it "caused" that to happen. It may simply be other confounding factors in the text, which are brought out by modifying that attribute, but the attribute itself (when causally traced, which we cannot currently do) had no effect.

In essence, it is more accurate to say "RATE measure the causal effect of **removing** an attribute on the reward", but not the attribute itself.

**Essential References Not Discussed:**

This is ok as far as I know.

**Experimental Designs Or Analyses:**

The experiments are fine for what the authors are trying to show.

**Methods And Evaluation Criteria:**

Yes, no issues here.

**Other Comments Or Suggestions:**

Line 76 typo

**Other Strengths And Weaknesses:**

### Strengths
* This is an important problem, we should ideally better understand reward models if we are chasing human-AI alignment.
* The authors have thought carefully about common issues in counterfactual generation and are trying to directly address them.

### Weaknesses
* Clarity was a bit of an issue for me. I found it very difficult to understand the process going from original -> edit -> edit of edit. Perhaps the other reviewers got it a bit better, but I feel the paper would really have benefited from a clear motivating example as "Figure 1" to really make the problem setup clear, and the issues with prior approaches, as the paper is I had to mentally work very hard to (I think) understand the setup. Figure 1 currently doesn't make it clear how edits can really help.
* It's difficult to judge Figure 2 also, is the difference in distributions significant? I believe the only way to really evaluate this is with pairwise comparisons to show a change/no change.
* At a deeper level, my my concern with the paper is that the authors don't show how the method is useful in application. What would make the paper much better in my eyes would be to demonstrate its usage for e.g. model improvement. Can you show it improves LLMs in practice? That would be the only truly compelling evaluation in my opinion.

**Questions For Authors:**

* Can you absolutely guarantee that the counterfactuals (and additional edits) form reasonable text? The method seems to depend upon the second edit just "minimally" editing the first edit? How do you know it does that? Creating a counterfactual from a counterfactual seems to multiply the issue of confounding factors rather than help? But maybe I just misunderstand.
* Can you add an evaluation showing your method helps with model improvement? (or anything else useful)

If you can address these convincingly, I will raise my score to accept.

**Relation To Broader Scientific Literature:**

Explainability of reward models is mostly a novel research area, so it does help to fill a gap in that sense. Work by Jiang et al. (2024) did just do something similar, but as far as I know those authors did not consider the idea of rewriting counterfactual edits.

**Theoretical Claims:**

Did not check.

---

> ### Author Rebuttal · Authors · 2025-03-31
>
> Thanks for your review. We agree that the behavior of reward models is an important and understudied area of alignment research.
>
> First, we will respond to some higher level points:
> - Regarding your concern about the causal claims, we think there may be a confusion here. We are not defining our counterfactual examples as examples that result in a changed classification decision, as in Jiang, et al. 2024. Rather, we are creating counterfactual examples with our rewrite process that differ on one binary attribute (and rewrite noise). That is, our counterfactuals are of the form “what would the text of this response be if it were generated with binary latent variable W set to 1 rather than 0.” We then use these counterfactual examples to estimate an ATE on the reward score itself. That is, RATE does not remove attributes, but rather changes the value of attributes (e.g., swapping sentiment from positive to negative) in order to understand how the reward model’s output is affected by the attribute. So, our treatment effect is in “units of reward score” regardless of any classification decision.  We agree that we should mention the Jiang work earlier and more clearly distinguish our notion of counterfactual from the notion of counterfactual explanations addressed in Jiang. We have updated the manuscript to clarify this in the introduction.
> - We agree that the ultimate goal of RATE is model improvement. However, the current alignment pipeline involves several different components. If an aligned model is producing undesirable behavior (e.g. excessively long responses, annoying bullet pointed lists, etc.) it’s not obvious where in the pipeline this behavior is coming from. Is this from the data used or generated during alignment? The reward model? An unintended consequence of the alignment objective? We focus on a modular evaluation of reward models themselves in order to localize failure points in the alignment procedure, and we plan to examine the impact of reward model bias on aligned models in future work.
>
> Here are responses to some other comments and questions:
> - Thank you for suggesting improving figure 1. We agree that this leaves something to be desired. Edit: Here is a link to the updated figure https://postimg.cc/9wwXZ1bQ
> - Thanks for catching the typo on line 76. This is fixed on the updated version of the paper.
> - Can you clarify what you mean by “significant” regarding the difference in distributions in figure 2? “Significant” as in hypothesis testing or as in “meaningful”?
> - It’s not possible to *guarantee* that the rewrite procedure produces sensible text, but examination of (many) randomly selected examples shows the procedure is robust. When applying the method, it is, as always, important for practitioners to spot-check the data.
> - Regarding your question about Assumption 1, note that Assumption 1 is actually much weaker than the $\xi$ terms cancelling out for each unit: rather, it suffices for the cancellations to occur in expectation, which is very permissive. This is easy to check via visual inspection on a random subset.

---

### Decision · Program_Chairs · 2025-05-01

**Decision:**

Accept (poster)

**Comment:**

Reviewers have raises important concerns regarding the assumptions made in the paper and the comparison to Jiang et al. 2024.
Reviewer qahC summarises the discussion on the paper very well. Without doubt, the reliance on the two assumptions limits the proposed method, as they are unlikely to hold in practice. This is the achilles heel of the paper. On the other hand, the reviewers agree that the problem being tackled in the paper is sufficiently novel, interesting and important. Also the the approach itself is clearly explained and simple to implement, with sound theoretical justification. Overall, the positive aspects outweigh the negative one.